# TP-Blend: Textual-Prompt Attention Pairing for Precise Object-Style Blending in Diffusion Models

**Xin Jin**  *felixxinjin@gmail.com*
*GenPi Inc.*

**Yichuan Zhong**  *yichuanzhong27@gmail.com*
*GenPi Inc.*

**Yapeng Tian**  *yapeng.tian@utdallas.edu*
*The University of Texas at Dallas*

Reviewed on OpenReview: *https://openreview.net/forum?id=q6M73uOBZE*

## Abstract

Current text-conditioned diffusion editors handle single object replacement well but struggle when a new object and a new style must be introduced simultaneously. We present Twin-Prompt Attention Blend (TP-Blend), a lightweight training-free framework that receives two separate textual prompts, one specifying a blend object and the other defining a target style, and injects both into a single denoising trajectory. TP-Blend is driven by two complementary attention processors. Cross-Attention Object Fusion (CAOF) first averages head-wise attention to locate spatial tokens that respond strongly to either prompt, then solves an entropy-regularised optimal transport problem that reassigns complete multi-head feature vectors to those positions. CAOF updates feature vectors at the full combined dimensionality of all heads (e.g., 640 dimensions in SD-XL), preserving rich cross-head correlations while keeping memory low. Self-Attention Style Fusion (SASF) injects style at every self-attention layer through Detail-Sensitive Instance Normalization. A lightweight one-dimensional Gaussian filter separates low- and high-frequency components; only the high-frequency residual is blended back, imprinting brush-stroke-level texture without disrupting global geometry. SASF further swaps the Key and Value matrices with those derived from the style prompt, enforcing context-aware texture modulation that remains independent of object fusion. Extensive experiments show that TP-Blend produces high-resolution, photo-realistic edits with precise control over both content and appearance, surpassing recent baselines in quantitative fidelity, perceptual quality, and inference speed.

## 1 Introduction

Text-driven image editing with diffusion models Brack et al. (2024); Brooks et al. (2023); Sheynin et al. (2024); Mokady et al. (2023); Liu et al. (2024); Tumanyan et al. (2023); Chen et al. (2024); Avrahami et al. (2023); Ge et al. (2023); Shi et al. (2024); Deutch et al. (2024); Li et al. (2024b) has excelled at tasks like object replacement but still lacks a robust solution for object blending, where two objects must fuse seamlessly into a single coherent entity. Achieving such morphological transitions is challenging: the system must preserve each source object's defining characteristics (e.g., color, shape, texture) while synthesizing intermediate attributes that accurately reflect the intended blend. This capability is especially valuable in creative design, film production, product prototyping, and scientific or educational visualization, where smooth transitions (e.g., morphing a car into a spaceship or combining organisms to study evolutionary traits) are often essential.

Most style transfer methods still rely on reference images, limiting users to existing examples and requiring substantial effort Chung et al. (2024); Xing et al. (2024); Wang et al. (2024a); Xu et al. (2024); Li (2024); Lötzsch et al. (2022); Wang et al. (2023). By contrast, text-driven approaches Hertz et al. (2024); Zhang et al. (2023); Liu et al. (2023); Wu et al. (2024a) specify styles in natural language (e.g., "sketch-like," "art nouveau") and could offer greater flexibility, yet they remain underexplored.

Additionally, current style transfer techniques face major obstacles in achieving fine-grained, multi-scale, and region-specific control. They often fail to capture high-frequency textural details, losing subtle stylistic cues (e.g., brushstrokes, grain, intricate material features) even at high resolutions Chung et al. (2024); Xing et al. (2024), thereby compromising overall texture fidelity.

Motivated by these challenges, we propose Twin-Prompt Attention Blend (TP-Blend), a training-free framework that extends Classifier-Free Guided Text Editing (CFG-TE) Brack et al. (2024); Brooks et al. (2023); Sheynin et al. (2024); Mokady et al. (2023); Liu et al. (2024); Tumanyan et al. (2023); Chen et al. (2024); Avrahami et al. (2023); Ge et al. (2023) to support fine-grained object blending and style fusion through separate textual prompts, as illustrated in Figure 1. TP-Blend introduces two new modules: Cross-Attention Object Fusion (CAOF), which integrates features from a blend object prompt using attention maps and an Optimal Transport framework; and Self-Attention Style Fusion (SASF), which injects style via Detail-Sensitive Instance Normalization (DSIN) and replaces self-attention Key/Value matrices with those from the style prompt. Unlike prior image-based approaches, TP-Blend enables direct textual control of both content and style, offering precise and independent modulation of blending strength and texture details. By unifying object replacement, blending, and style transfer within a single denoising process, TP-Blend enhances controllability without incurring additional computational overhead.

**Main Contributions.** (1) Dual-Prompt Mechanism decouples object and style prompts, preventing interference and ensuring precise content representation and faithful style transfer within a unified denoising process; (2) CAOF with Optimal Transport aligns and integrates blend-object features into a replaced object by treating attention maps as distributions, enabling seamless morphological transitions and preserving semantic integrity; (3) SASF leverages DSIN to extract and transfer high-frequency style features, preserving intricate textural details without over-smoothing while allowing adaptive modulation of stylistic attributes across different spatial extents and granularities; (4) text-driven Key/Value substitution replaces self-attention Key/Value matrices with those derived from the style prompt, enforcing localized style modulation while maintaining spatial coherence and object fidelity.

**Code Availability.** Code is available at `https://github.com/felixxinjin1/TP-Blend`.

## 2 Related Work

Diffusion models have emerged as the dominant paradigm for text-guided image generation and editing, beginning with unconditional DDPMs and latent variants together with classifier-free guidance, which underpin early editing systems but leave open challenges in multi-concept disentanglement and precise regional control (Dhariwal & Nichol, 2021; Rombach et al., 2022; Ho & Salimans, 2022; Brooks et al., 2023; Brack et al., 2024). Unified generation and editing continues to broaden capability and instruction following, spanning single or dual diffusion formulations, universal dynamics-aware frameworks, adapter-based unification, and interactive or conversational editors that couple instruction tuning with in-context visual reasoning (Liu et al., 2025d; Yu et al., 2024; 2025a; Xia et al., 2024; Zhou et al., 2025b; Xia et al., 2025; Xiao et al., 2025b; Chen et al., 2025d; Duan et al., 2025; Fu et al., 2025b; Li et al., 2025h; Jia et al., 2025a; Lee et al., 2025b; Chen et al., 2025c; Xu et al., 2025; Zhou et al., 2025a; Li et al., 2025g; Wu et al., 2025d; Jia et al., 2025b; Lai et al., 2025; Mao et al., 2025a; Cao et al., 2025; Chen et al., 2025b; Luo et al., 2025a). Compositionality, multi-conditioning and regional control are advanced by balancing or transplanting representations, ordering and counting constraints, language-guided tokenization and unified tokenizers, as well as training-free or attention-repositioning guidance that improves spatial faithfulness and occlusion handling (Luo et al., 2025b; Jin et al., 2025; Cohen et al., 2024; 2025; Liang et al., 2025; Li et al., 2025d; Binyamin et al., 2025; Zeng et al., 2025; Hsiao et al., 2025; Qiu et al., 2025; Han et al., 2025c; Zhan & Liu, 2025; Wang et al., 2024b; Zha et al., 2025; Qu et al., 2025b; Wu et al., 2024b). Personalization and identity preservation leverage feature

| Original Image | Object Replacement | Object Blending | Style Blending |
|:---:|:---:|:---:|:---:|

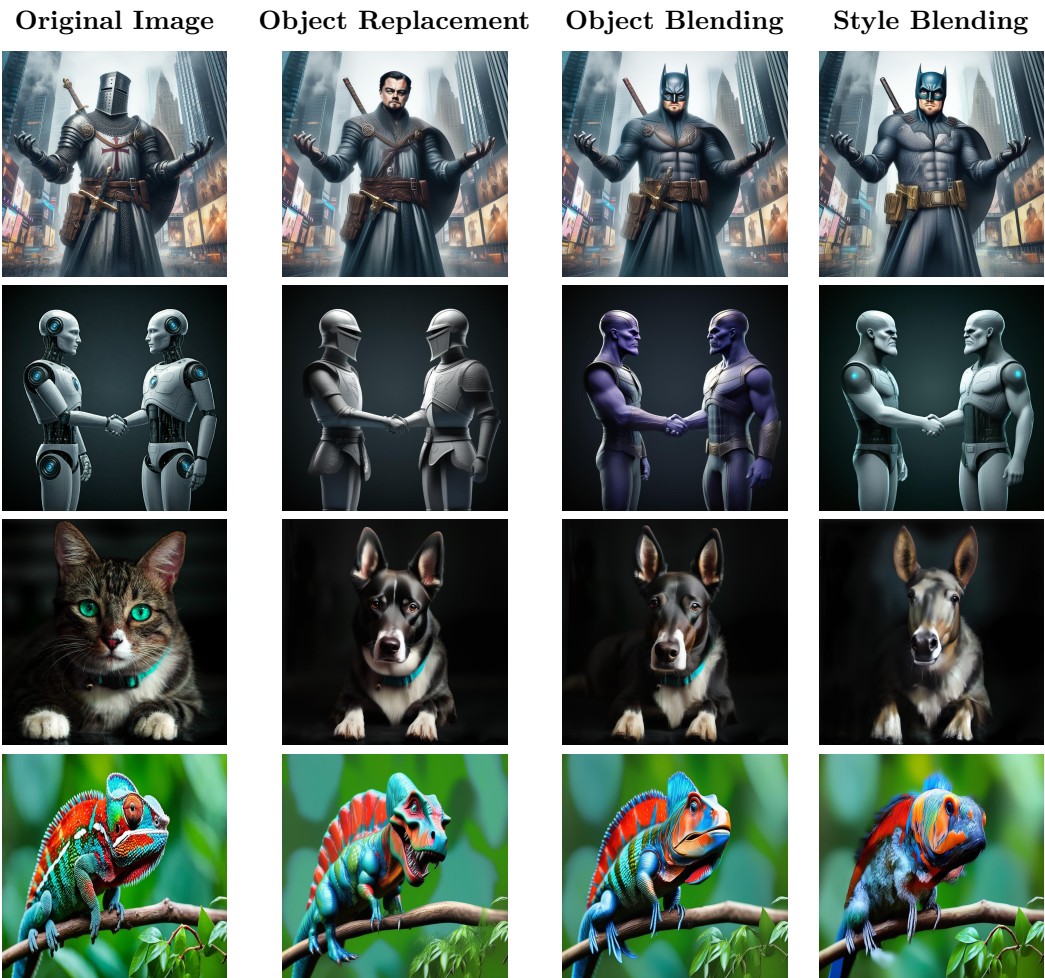

Figure 1: Demonstration of our method's capabilities. Row 1: Original object "**Knight**" is replaced by "**Leonardo DiCaprio**", blended with "**Batman**", and styled with "**Pop Art**". Row 2: Original object "**Robot**" is replaced by "**Knight**", blended with "**Thanos**", and styled with "**Cyberpunk Style**". **Row 3**: Original object "**Cat**" is replaced by "**Dog**", blended with "**Horse**", and styled with "**Oil painting**". **Row 4**: Original object "**Chameleon**" is replaced by "**Dinosaur**", blended with "**Fish**", and styled with "**Oil painting**".

caching and standardize-then-personalize pipelines, continual concept learning with mitigation of forgetting and confusion, parameter-efficient or time-aware frequency guidance, consistency from a single prompt and localized attention, LoRA merging and hypernetworks, and targeted human preference optimization (Aiello et al., 2025; Xie et al., 2025a; Guo & Jin, 2025; Li et al., 2025c; Liu et al., 2025c; Huang et al., 2025c; Liu et al., 2025e; Xiao et al., 2025a; Shenaj et al., 2024; Na et al., 2025; Zhu et al., 2025a; Li et al., 2025e; Wang et al., 2025d; Liu et al., 2025f). Style control has moved from exemplar dependence toward text-driven and detail-faithful modulation through state-space modeling, selective style element control, object-centric style editing, causal intervention and conflict-free guidance, including emotional manipulation (Liu et al., 2025a; Lei et al., 2025; Park et al., 2025b; Huang et al., 2025e; Jo et al., 2025; Yang et al., 2025d; Cai et al., 2024a; Huang et al., 2025d; Dang et al., 2025). Efficiency and scheduling improve with learned time prediction, one-step distillation and time-independent encoders, few-step or lightning editors, second-order sampling, fast inpainting and latent super-resolution, together with scale-down text encoders (Ye et al., 2024; 2025; Luo et al., 2024; Nguyen et al., 2024; 2025b;a; Chadebec et al., 2025; Wang et al., 2025a; Xie et al., 2025b; Jeong et al., 2025; Li et al., 2025e; Wang et al., 2025c). Scaling and backbone design target long-range coherence and high resolution via hierarchical transformers, frequency or attention modulation and resolution-agnostic

diffusion, while autoregressive alternatives push tokenization and decoding with large tokenizers, holistic or language-guided tokenizers, deep-compression hybrids and frequency-aware modeling (Zhang et al., 2025a; Voronov et al., 2024; Yellapragada et al., 2024; Yang et al., 2025b; Shi et al., 2025b; Wang et al., 2025b; Han et al., 2025b; Kumbong et al., 2025; Han et al., 2025a; Wu et al., 2025b; Xiong et al., 2025; Zheng et al., 2025a; Zha et al., 2025; Qu et al., 2025b; Wu et al., 2025e; Chen et al., 2025e; Zheng et al., 2025b; Qu et al., 2025a; So et al., 2025). Domain and structure-aware generation spans design and line-art images, creative and degraded layouts with cycle consistency, unified layout planning, grounded instance-level control and multi-view consistency, scene-consistent camera control, controllable street views, variable multi-layer transparency and layer decomposition, and decoupled inter and intra element conditions (Wang et al., 2025g; 2024c; Zhang et al., 2024; Cai et al., 2024b; He et al., 2025; Wang et al., 2025e; Wu et al., 2024b; Huang et al., 2024; Yuan et al., 2025; Gu et al., 2025; Pu et al., 2025; Yang et al., 2025e;c; Wu et al., 2025a). Safety, fairness and provenance are studied through typographic threats, trigger-free branding attacks, robust watermarking, anti-editing and localized concept erasure, guideline-token safety, fair mapping and debiasing with prompt optimization or entanglement-free attention, along with human-centric quality metrics (Cheng et al., 2025; Jang et al., 2025; Wang et al., 2025i;h; Lee et al., 2025a; Li et al., 2025a;b; Um & Ye, 2025; Kim et al., 2025b; Park et al., 2025a; Wang et al., 2025f; Huang et al., 2025b). Evaluation and analysis expand with enhanced compositional benchmarks, scalable human-aligned editing assessment, trade-off studies and probabilistic analyses, and work on paragraph-level grounding and attribute-centric composition (Huang et al., 2025a; Ryu et al., 2025; Zhang et al., 2025d; Yu et al., 2025b; Cong et al., 2025; Wu et al., 2025d; Xuan et al., 2025; Zhu et al., 2025b;c; Ren et al., 2025). Beyond images, image-to-video and motion-controlled generation integrate text-to-image with text-to-video models or decouple motion intensity for one-step or high-quality synthesis (Shi et al., 2025a; Mao et al., 2025b; Su et al., 2025). Practical deployment advances include mobile-oriented high-resolution generation (Chen et al., 2025a). Additional controls include quantity perception, dynamic-feedback regulation, region-aware fine-tuning, collaborative attention, proportional group representations, zero-shot multi-attribute creation, effective and diverse prompt sampling, layer-wise memory, semantic-faithful noise diffusion and spatial attention repositioning, plus zero-shot subject-driven generation by large inpainting models (Li et al., 2025f; Fu et al., 2025a; Xing et al., 2025; Yang et al., 2025a; Jung et al., 2025; Deng et al., 2025; Yun et al., 2025; Kim et al., 2025a; Miao et al., 2025; Han et al., 2025c; Shin et al., 2025). Integration with large language models supports planning, reward shaping and stepwise verification through collaborative or customized multimodal rewards, chain-of-thought reinforcement and preference optimization for alignment and controllability (Liu et al., 2025b; Tang et al., 2025; Ba et al., 2025; Zhou et al., 2025c; Jiang et al., 2025; Zhang et al., 2025b; Sun et al., 2025; Wu et al., 2025c; Guo et al., 2025; Fang et al., 2025). Complementary representations and training-free guidance broaden the modeling space with triplanes, rectified flows, dense-aligned guidance, Stable Flow and 3D-aware scene modeling, together with hyperbolic diffusion autoencoders, unified self-supervised pretraining and industrial anomaly generation (Bilecen et al., 2024; Dalva et al., 2024; Wang et al., 2025j; Avrahami et al., 2024; 2025; Zhang et al., 2025c; Li et al., 2024a; Chu et al., 2025; Dai et al., 2024; Cai et al., 2025b). Our work is situated within these trends while retaining links to foundational and domain-specific efforts such as Design-Diffusion, LineArt, Focus-N-Fix, Type-R and PreciseCam and contemporaneous advances in layout, control and training-free customization (Wang et al., 2025g; 2024c; Xing et al., 2025; Shimoda et al., 2024; Bernal-Berdun et al., 2025; Zhang et al., 2025a; Cai et al., 2025a). For completeness, we note additional concurrent directions spanning instruction-following editors, multimodal integration, compositional sliders and broader analyses (Cao et al., 2025; Chen et al., 2025b; Luo et al., 2025a; Qu et al., 2025a; So et al., 2025; Ren et al., 2025; Zhu et al., 2025c;b).

## 3 Proposed Method

### 3.1 Preliminaries

Diffusion models Dhariwal & Nichol (2021); Rombach et al. (2022) progressively denoise a latent variable to produce high-fidelity images. Classifier-Free Guidance (CFG) Ho & Salimans (2022) steers generation toward conditioning inputs by interpolating between conditional and unconditional noise predictions. Specifically, the model is trained to predict both $\boldsymbol{\epsilon}_\theta(\mathbf{x}_t)$ and $\boldsymbol{\epsilon}_\theta(\mathbf{x}_t, c)$, where $c$ is the conditioning. At inference, a guidance

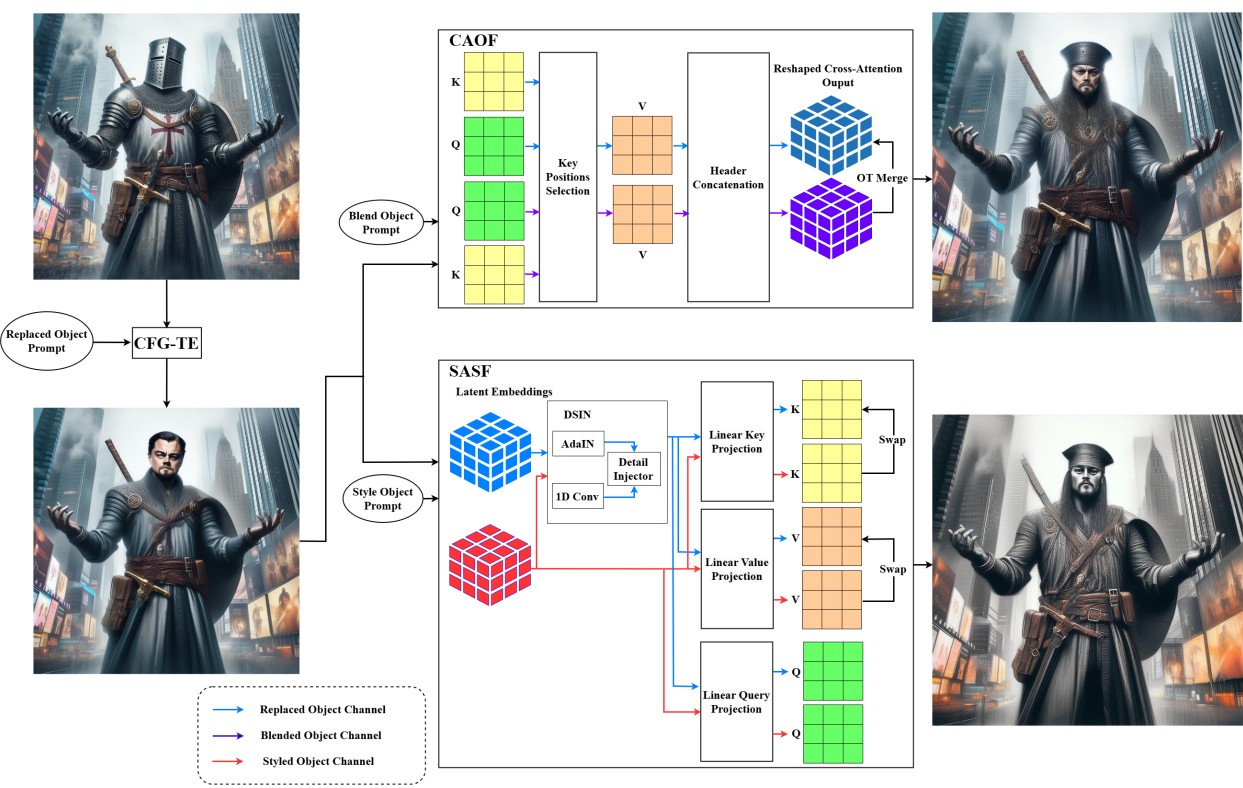

Figure 2: Flowchart of TP-Blend, integrating object replacement, blending, and style transfer within the diffusion process. In this example, the original object "**Knight**" is replaced by "**Leonardo DiCaprio**", blended with "**Captain Jack Sparrow**", and styled with a "**Charcoal Drawing**" effect.

scale $s_g$ modifies the predicted noise:

$$\tilde{\epsilon}_\theta(\mathbf{x}_t) \;=\; \epsilon_\theta(\mathbf{x}_t) \;+\; s_g\big(\epsilon_\theta(\mathbf{x}_t, c) - \epsilon_\theta(\mathbf{x}_t)\big). \tag{1}$$

CFG-TE extends CFG to perform precise edits on an existing image $\mathbf{x}_0$. The image is inverted to a latent $\mathbf{x}_T$ via DDIM inversion Song et al. (2020), which defines a deterministic mapping between timesteps when no guidance is applied:

$$\mathbf{x}_{t-1} \;=\; \sqrt{\alpha_{t-1}} \left( \frac{\mathbf{x}_t - \sqrt{1-\alpha_t}\,\epsilon_\theta(\mathbf{x}_t)}{\sqrt{\alpha_t}} \right) \;+\; \sqrt{1-\alpha_{t-1}}\,\epsilon_\theta(\mathbf{x}_t), \tag{2}$$

where $\alpha_t$ is the noise schedule. Once inverted, the noise prediction at each denoising step can be modified to remove or add concepts:

$$\tilde{\epsilon}_\theta^{\text{edit}}(\mathbf{x}_t) \;=\; \epsilon_\theta(\mathbf{x}_t) \;+\; s_e\,\Delta\epsilon_\theta(\mathbf{x}_t), \tag{3}$$

with

$$\Delta\epsilon_\theta(\mathbf{x}_t) = \begin{cases} \epsilon_\theta(\mathbf{x}_t, c_{\text{edit}}) - \epsilon_\theta(\mathbf{x}_t), & \text{(positive guidance)}, \\ \epsilon_\theta(\mathbf{x}_t) - \epsilon_\theta(\mathbf{x}_t, c_{\text{edit}}), & \text{(negative guidance)}, \end{cases} \tag{4}$$

where $s_e$ is the edit guidance scale and $c_{\text{edit}}$ is the editing prompt.

### 3.2 Twin-Prompt Attention Blend

CFG-TE enables object replacement by applying positive guidance to the new object prompt and negative guidance to the original. However, it lacks mechanisms for fine-grained *object blending* and *style fusion*, which require compositional mixing and textural transformations.

We introduce TP-Blend, extending CFG-TE with two additional prompts: a blend prompt and a style prompt, both assigned zero edit guidance to avoid interfering with object replacement. Cross-Attention Object Fusion (CAOF) integrates blend object features at key spatial positions using a unified attention map and an Optimal Transport framework. Self-Attention Style Fusion (SASF) modulates texture and style by locally adjusting feature statistics using DSIN and substituting the Key/Value matrices with those derived from the style prompt. By decoupling both blending and style transfer from the editing guidance scale, TP-Blend integrates seamlessly into the denoising process, enhancing CFG-TE's capabilities with high-fidelity object and style blending (Fig. 2).

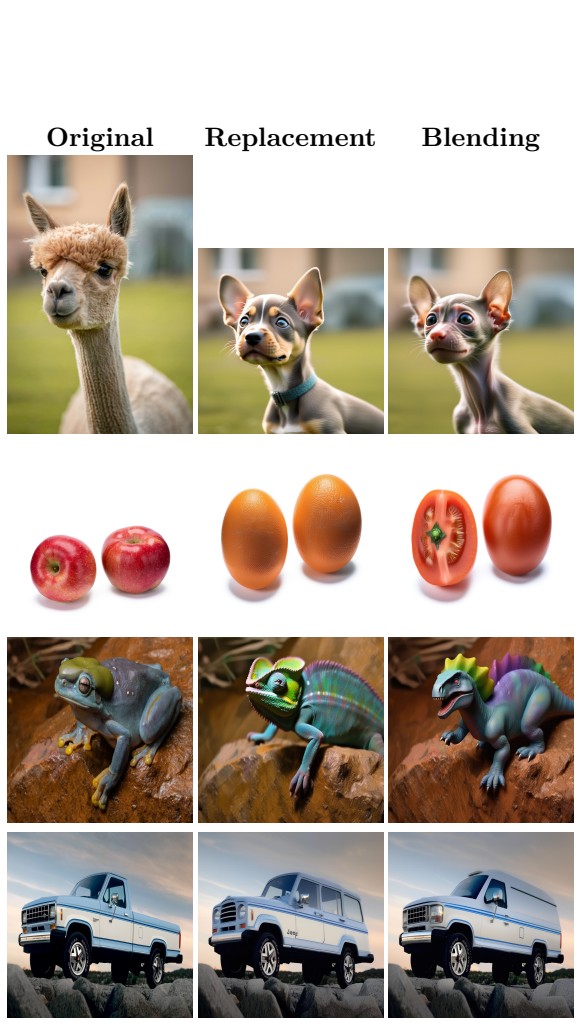

Figure 3: CAOF Object Blending across different sets. Row 1: Original object "**alpaca**" is replaced by "**puppy**" and blended with "**monkey**". Row 2: Original "**apple**" is replaced by "**orange**" and blended with "**tomato**". Row 3: Original "**frog**" is replaced by "**chameleon**" and blended with "**dinosaur**". **Row 4**: Original "**truck**" is replaced by "**jeep**" and blended with "**ambulance**".

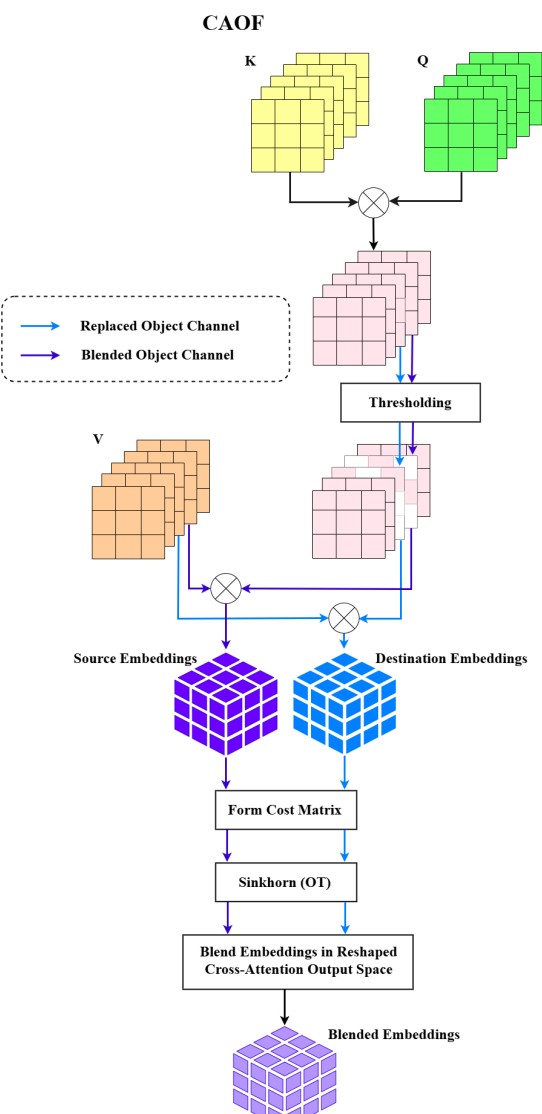

Figure 4: CAOF Flowchart: Cross-Attention Object Fusion merges the blend object's features into the replaced object by identifying key spatial positions in the attention maps and applying an optimal transport framework for coherent morphological transitions.

### 3.3 Cross-Attention Object Fusion

As shown in Figure 3 and summarized in Figure 4, CAOF seamlessly integrates a blend object's features into a replaced object during the diffusion process. Leveraging textual prompts for both the replaced and blend objects, CAOF locates key spatial regions in cross-attention maps and employs an Optimal Transport (OT) framework to determine blending levels.

**Identifying Significant Positions in Cross-Attention Maps.**  In multi-head cross-attention Vaswani (2017), each head $h$ produces attention weights

$$\mathbf{A}^{(h)} = \text{softmax}\left( \frac{\mathbf{Q}^{(h)} \mathbf{K}^{(h)\top}}{\sqrt{d_k}} \right), \tag{5}$$

where $\mathbf{Q}^{(h)} \in \mathbb{R}^{N \times d_k}$ and $\mathbf{K}^{(h)} \in \mathbb{R}^{M \times d_k}$ are query/key matrices, $N$ is the number of spatial positions, $M$ is the number of text tokens, and $d_k$ is the head dimension. We average over $H$ heads and focus on the replaced and blend object tokens, $t_{\text{replaced}}$ and $t_{\text{blend}}$:

$$\mathbf{a}_{\text{replaced}} = \frac{1}{H} \sum_{h=1}^{H} \mathbf{A}^{(h)}_{:,\, t_{\text{replaced}}}, \quad \mathbf{a}_{\text{blend}} = \frac{1}{H} \sum_{h=1}^{H} \mathbf{A}^{(h)}_{:,\, t_{\text{blend}}}. \tag{6}$$

To identify meaningful spatial positions, we introduce two percentile thresholds, $\tau_{\text{source}}$ and $\tau_{\text{dest}}$. Specifically, any position $i$ in $\mathbf{a}_{\text{blend}}$ whose attention weight exceeds the $\tau_{\text{source}}$-percentile is included in the source set $\mathcal{S}$, and any position in $\mathbf{a}_{\text{replaced}}$ exceeding the $\tau_{\text{dest}}$-percentile is placed in the destination set $\mathcal{D}$.

*Clarification.* The thresholds act on different head-averaged maps and therefore induce the two index sets independently. We make this explicit with

$$\mathcal{S} = \left\{ i \in [N] : \mathbf{a}_{\text{blend}}[i] \geq q_{\tau_{\text{source}}}(\mathbf{a}_{\text{blend}}) \right\}, \qquad \mathcal{D} = \left\{ i \in [N] : \mathbf{a}_{\text{replaced}}[i] \geq q_{\tau_{\text{dest}}}(\mathbf{a}_{\text{replaced}}) \right\}, \tag{7}$$

where $q_\tau(\cdot)$ denotes the $\tau$-percentile. The sets $\mathcal{S}$ and $\mathcal{D}$ are not a partition of the image tokens. A position can be in neither set when it is below both cutoffs, in exactly one set when it responds strongly to only one prompt, or in both sets when it responds strongly to both prompts. In the fusion step (Sec. 3.3), only destination positions $d \in \mathcal{D}$ are updated and they receive transported features from sources $s \in \mathcal{S}$ under the OT plan $\mathbf{T}$ (Eq. 11 and Eq. 9). Tokens $i \notin \mathcal{D}$ pass through unchanged. If a spatial location lies in $\mathcal{S} \cap \mathcal{D}$, its destination slot taken from the replaced stream can still import features from its source slot taken from the blend stream because these embeddings originate from different prompt branches which avoids ambiguity. When an object phrase spans multiple text tokens, we pool their columns before thresholding and use the mean by default, while max pooling yields similar behavior in our experiments. Although we often tie the thresholds in a joint setting with $\tau_{\text{source}} = \tau_{\text{dest}}$ to simplify usage (see Fig. 12), they are defined separately and can be chosen differently to trade precision and coverage. The percentile formulation makes the selection scale-free and robust across layers and prompts because it depends on rank within each map rather than absolute magnitude.

**Blending Feature Embeddings in Reshaped Cross-Attention Outputs.**  To effectively integrate features from the blend object into the replaced object, we begin by concatenating the per-head attention outputs along the feature dimension:

$$\mathbf{O} = \text{Concat}_{h=1}^{H}\left( \mathbf{A}^{(h)} \mathbf{V}^{(h)} \right) \in \mathbb{R}^{N \times D}, \tag{8}$$

where $\mathbf{A}^{(h)} \in \mathbb{R}^{N \times M}$ are attention weight matrices, $\mathbf{V}^{(h)} \in \mathbb{R}^{M \times d_k}$ are the corresponding value matrices, $D = H \cdot d_k$ is the total feature dimensionality, $N$ is the number of query positions, and $M$ is the number of key tokens. By consolidating multi-head outputs into a single representation, we preserve all information necessary for seamless fusion, avoiding the loss that would occur from per-head embeddings.

We then blend the feature vectors of the replaced object with those of the blend object under a transport plan $\mathbf{T}$. Specifically, if $d_i \in \mathcal{D}$ and $s_j \in \mathcal{S}$ denote destination and source positions respectively, with $\mathbf{f}_{d_i}, \mathbf{f}_{s_j} \in \mathbb{R}^D$

being their respective feature vectors from $\mathbf{O}$, the updated feature vector at position $d_i$ becomes

$$\mathbf{f}'_{d_i} = (1 - w_0)\,\mathbf{f}_{d_i} + w_0 \sum_{s_j \in \mathcal{S}} \frac{T_{ij}}{\sum_{s_k \in \mathcal{S}} T_{ik}}\,\mathbf{f}_{s_j}, \tag{9}$$

where $w_0 \in [0, 1]$ controls the relative influence of the blend features, and $T_{ij}$ is obtained by solving the **OT problem**. By treating the multi-head outputs as a whole at the full dimensionality (e.g., $D = 640$), we not only preserve complex content and style cues but also obtain a more manageable OT cost matrix (e.g., $4096 \times 4096$), avoiding the significantly larger matrices (e.g., $40960 \times 40960$) that would result from per-head processing.

**Formulating the Optimal Transport Problem.** Let $\mathcal{S}$ and $\mathcal{D}$ denote the sets of source (blend object) and destination (replaced object) positions. The cost of transporting mass from source position $j \in \mathcal{S}$ to destination position $i \in \mathcal{D}$ is given by

$$C_{ij} = \lambda_{\text{feature}}\,D_{\text{feature}}(i, j) + \lambda_{\text{spatial}}\,D_{\text{spatial}}(i, j), \tag{10}$$

where $D_{\text{feature}}(i, j)$ is the cosine distance between feature vectors $\mathbf{f}_i$ and $\mathbf{f}_j$ and $D_{\text{spatial}}(i, j)$ is the Euclidean distance between their spatial coordinates.

We solve the entropic OT problem:

$$\min_{\mathbf{T} \geq 0} \quad \sum_{i \in \mathcal{D}} \sum_{j \in \mathcal{S}} T_{ij} C_{ij} - \gamma H(\mathbf{T}), \tag{11}$$

$$\text{s.t.} \quad \sum_{j \in \mathcal{S}} T_{ij} = 1, \quad \forall i \in \mathcal{D}, \tag{12}$$

$$\sum_{i \in \mathcal{D}} T_{ij} \geq \frac{1}{|\mathcal{S}|}, \quad \forall j \in \mathcal{S}, \tag{13}$$

where $H(\mathbf{T}) = -\sum_{i,j} T_{ij} \log T_{ij}$ is the entropy term, and $\gamma > 0$ is the regularization parameter. Entropy regularization promotes smoother transport mass across source-destination pairs.

**Solving the Optimal Transport Problem with the Sinkhorn Algorithm.** The entropic regularization allows the problem to be efficiently solved using the Sinkhorn algorithm Cuturi (2013); Peyré et al. (2019); Genevay et al. (2016). We form the Gibbs kernel $\mathbf{K} = \exp(-\mathbf{C}/\gamma)$ and iteratively update scaling vectors $\mathbf{u} \in \mathbb{R}^{|\mathcal{D}|}$ and $\mathbf{v} \in \mathbb{R}^{|\mathcal{S}|}$:

$$\mathbf{u}^{(k+1)} = \frac{\mathbf{1}_{|\mathcal{D}|}}{\mathbf{K}\,\mathbf{v}^{(k)}}, \quad \mathbf{v}^{(k+1)} = \frac{\frac{1}{|\mathcal{S}|}\,\mathbf{1}_{|\mathcal{S}|}}{\mathbf{K}^{\top}\,\mathbf{u}^{(k+1)}}, \tag{14}$$

until convergence. The transport plan becomes

$$\mathbf{T} = \text{diag}(\mathbf{u})\,\mathbf{K}\,\text{diag}(\mathbf{v}). \tag{15}$$

Finally, we use $\mathbf{T}$ to blend each destination feature with weighted contributions from the source. Reintegrating these blended features into the cross-attention outputs yields a naturally fused object that inherits characteristics of the blend object at selected positions, with minimal overhead or artifacts.

### 3.4 Self-Attention Style Fusion

As illustrated in Figure 5 and outlined in Figure 6, SASF integrates style and texture into the replaced object through self-attention. Compared to previous methods Chung et al. (2024); Xing et al. (2024); Wang et al. (2024a); Xu et al. (2024); Li (2024); Lötzsch et al. (2022); Hertz et al. (2024); Zhang et al. (2023); Wang et al. (2023), SASF offers four advantages: (1) it introduces DSIN to capture HF textural details in a lightweight yet effective manner; (2) it relies on simple textual prompts rather than style images; (3) it fuses style and object features simultaneously during denoising, preserving both content fidelity and style

coherence; and (4) By translating historical idioms such as *Ukiyo-e*, *Renaissance*, and *Baroque* into their own material vocabularies, SASF can recode fabric weave, ornamentation, and weaponry; chain mail shifts to brocaded velvet, a plain sword strap becomes an obi sash, yet the figure's stance and the surrounding cityscape stay unchanged, as demonstrated in Figure 7.

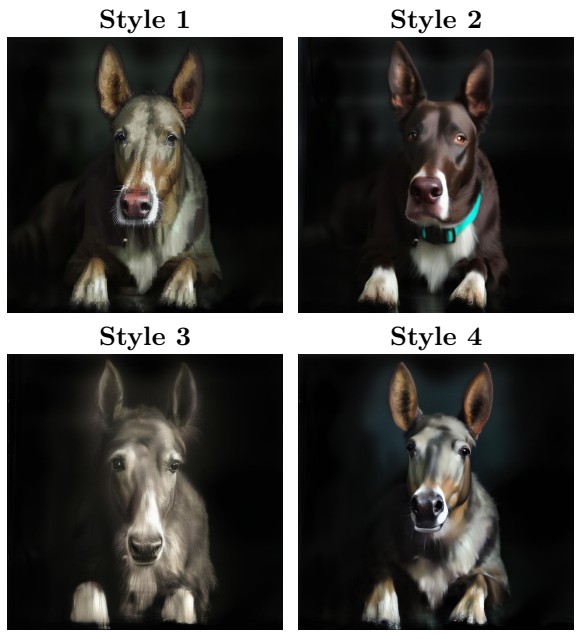

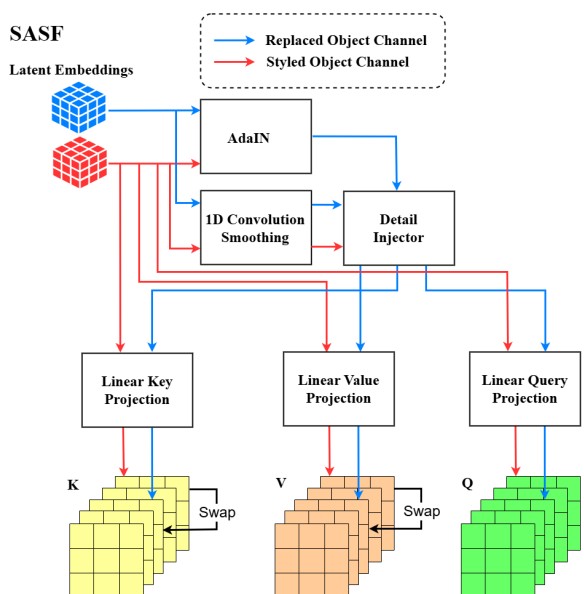

Figure 5: Object blending enhanced with various artistic styles. **Style 1**: Pixel Art; **Style 2**: Chocolate; **Style 3**: Charcoal Drawing; **Style 4**: Oil Painting.

Figure 6: SASF Flowchart: Self-Attention Style Fusion incorporates style prompts by injecting high-frequency details via DSIN and substituting textual Key/Value matrices, ensuring fine-grained style modulation during the diffusion process.

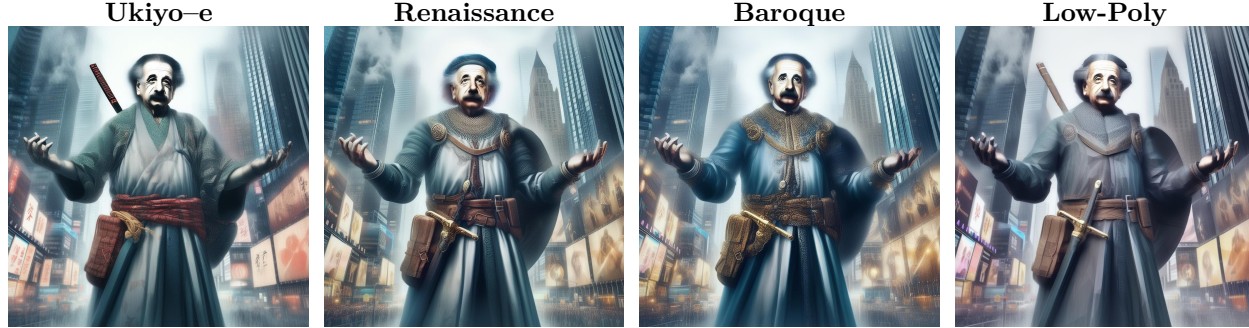

Figure 7: **Stylistic renderings reshape fabric texture and accessories while pose and setting remain unchanged.** The original **knight** is replaced by **Albert Einstein**, blended with a **nobleman** concept, and then rendered in four distinct styles. Each style reinterprets the garments in a unique way: **Ukiyo-e** replaces the surcoat with a patterned kimono, complete with an obi sash and a lacquered katana; **Renaissance** introduces brocaded velvet, gilt medallions, and a scholar's cap; **Baroque** presents deep hued silk enriched with heavy gold embroidery and filigreed weaponry; **Low-Poly** abstracts every surface into planar facets and simplifies folds and metallic highlights.

**Detail-Sensitive Instance Normalization.** Let $\mathbf{F}_{\text{replaced}}, \mathbf{F}_{\text{style}} \in \mathbb{R}^{N \times D}$ be the latent embeddings (i.e., token-wise feature maps) of the replaced and style objects, respectively. We first perform an AdaIN step on

the replaced features:

$$\mathbf{F}'_{\text{replaced}} = \left( \frac{\mathbf{F}_{\text{replaced}} - \mu_{\text{rep}}}{\sigma_{\text{rep}}} \right) \sigma_{\text{style}} + \mu_{\text{style}}, \tag{16}$$

where $(\mu_{\text{rep}}, \sigma_{\text{rep}})$ and $(\mu_{\text{style}}, \sigma_{\text{style}})$ are the channel-wise means and standard deviations of the replaced and style embeddings. This aligns global statistics (mean and variance) to match the target style, but by itself may overlook subtle, higher-frequency stylistic cues.

Next, DSIN applies a small 1D Gaussian smoothing filter along the token dimension to decompose both $\mathbf{F}_{\text{replaced}}$ and $\mathbf{F}_{\text{style}}$ into low-frequency (LF) and high-frequency (HF) components:

$$\mathbf{F}^{\text{LF}} = \mathbf{F} * \mathbf{K}, \quad \mathbf{F}^{\text{HF}} = \mathbf{F} - \mathbf{F}^{\text{LF}}, \tag{17}$$

where $\mathbf{K}$ is a 1D Gaussian kernel of size $k = 2m + 1$ and width $\sigma$. Intuitively, $\mathbf{F}^{\text{LF}}$ captures coarse variations (slower changes across tokens), while $\mathbf{F}^{\text{HF}}$ isolates the finer details. DSIN then injects a fraction $\alpha$ of the style HF difference directly into the AdaIN output:

$$\mathbf{F}''_{\text{replaced}} = \mathbf{F}'_{\text{replaced}} + \alpha \left( \mathbf{F}^{\text{HF}}_{\text{style}} - \mathbf{F}^{\text{HF}}_{\text{replaced}} \right). \tag{18}$$

When DSIN applies a 1D Gaussian kernel $\mathbf{K}$ along the token dimension, it acts as a low-pass filter in the frequency domain: larger $\sigma$ broadens the kernel's passband, yielding a narrower high-frequency (HF) residual $\mathbf{F}^{\text{HF}}$ and thus a subtler style injection. Conversely, smaller $\sigma$ captures more mid- and high-frequency components, accentuating textural details (e.g., brushstrokes) in the final output. The injection fraction $\alpha$ then scales the amplitude of these style-specific HF cues. In effect, $\sigma$ and $\alpha$ together provide a powerful mechanism for tuning the granularity and prominence of style features.

Unlike prior approaches such as Huang & Belongie (2017) or Chung et al. (2024) that apply AdaIN globally or only at the initial noise level for DDIM inversion, our DSIN is applied at *every self-attention layer* throughout the denoising process. This repeated application ensures the progressive and layer-wise infusion of fine-grained stylistic features, enabling multi-scale texture adaptation without disrupting the overall structure.

**Key/Value Substitution.** Following the DSIN framework, we first construct the Query, Key, and Value matrices for self-attention. We then substitute the Key and Value channels of the target (replaced) object with those of the style source:

$$\mathbf{K}_{\text{tar}} \leftarrow \mathbf{K}_{\text{sty}}, \qquad \mathbf{V}_{\text{tar}} \leftarrow \mathbf{V}_{\text{sty}}. \tag{19}$$

Since the self-attention output is computed by weighting the Value vectors using Query-Key dot products, replacing the Key and Value matrices of the replaced region with those from the style prompt allows style features to dominate the attention updates. This substitution imposes the texture and local patterns of the style onto the replaced object, leading to strong stylistic transformations.

While Chung et al. (2024) apply this substitution using Key/Value representations extracted from an image-based style encoder, our approach instead derives these from textual prompts. Specifically, we construct the Key/Value matrices from the text prompts of both the replaced object and the style source, enabling a text-driven style transfer mechanism without requiring image-based features.

Importantly, although this substitution offsets the effect of DSIN modulation in the Key/Value branches for the replaced object (since it is overwritten by style-derived features), DSIN-modified features remain intact in the Query branch. This asymmetry allows DSIN to still influence the attention outputs via its role in computing attention scores. Consequently, high-frequency stylistic cues injected through DSIN continue to impact the hidden embeddings passed to the next layer. This achieves a dual effect: the Key/Value substitution enforces stylistic consistency, while DSIN-enhanced Queries preserve the structural fidelity of the replaced object, allowing nuanced and locally-aware style transfer.

Table 1: Performance on the Object Replacement + Object Blending task.

| Method | BOM↑ | $\text{CLIP}_R$↑ | $\text{CLIP}_B$↑ | $1-\text{LPIPS}_O$↑ |
|---|---|---|---|---|
| IP2P Brooks et al. (2023) (CVPR 2023) | 0.1075 | 0.1819 | 0.2708 | 0.5887 |
| StyleAligned Hertz et al. (2024) (CVPR 2024) | 0.2371 | 0.2120 | 0.2866 | 0.5814 |
| TurboEdit Deutch et al. (2024) | 0.3199 | 0.1984 | 0.2781 | 0.6125 |
| FLUX.1 Kontext Batifol et al. (2025) | 0.3401 | 0.2013 | 0.2898 | 0.6007 |
| LEDITS++ Brack et al. (2024) (CVPR 2024) | 0.3913 | 0.2078 | 0.2834 | 0.6145 |
| SeedEdit Shi et al. (2024) | 0.5486 | 0.2096 | 0.2966 | 0.6381 |
| Step1X-Edit Liu et al. (2025d) | 0.7150 | 0.2120 | 0.2913 | 0.7024 |
| Blended diffusion Avrahami et al. (2022) (CVPR 2022) | 0.7241 | 0.2026 | 0.2910 | 0.7568 |
| Nano Banana | 0.7324 | 0.2159 | 0.2866 | 0.7303 |
| **CAOF** | **0.8031** | 0.2014 | 0.2937 | 0.8292 |

## 4 Experiments

### 4.1 Implementation Details

**Model Architecture.** All experiments employ SD-XL Podell et al. (2023) as the diffusion backbone. The source image is first inverted to a latent $\mathbf{x}_T$ via DDIM inversion, guaranteeing exact reconstruction before editing. During the forward denoising pass we apply, at every timestep: (i) TIE-CFG for object replacement (positive guidance on the target prompt, negative on the original); (ii) CAOF to transport blend-object features into attention positions selected by the joint percentile thresholds $\tau_{\text{source}} = \tau_{\text{dest}} \in \{0.6, 0.7\}$; and (iii) SASF to inject style via DSIN and key-value substitution. The Sinkhorn regulariser is fixed to $\gamma = 0.1$, with cost weights $\lambda_{\text{feature}} = 0.7$ and $\lambda_{\text{spatial}} = 0.3$ (Eq. 10).

**Baseline Methods.** To isolate the contribution of TP-Blend, we compare against six state-of-the-art text-driven editors and re-tune their prompts for each task so that every method receives semantically equivalent conditioning. The baselines are Step1X-Edit Liu et al. (2025d), SeedEdit Shi et al. (2024), LEDITS++ Brack et al. (2024) (CVPR 2024), StyleAligned Hertz et al. (2024) (CVPR 2024), TurboEdit Deutch et al. (2024), IP2P Brooks et al. (2023),(CVPR 2023), Blended diffusion Avrahami et al. (2022),(CVPR 2022), FLUX.1 Kontext Batifol et al. (2025), and Nano Banana.

SeedEdit and Step1X-Edit are inversion-free decoders optimised for speed, LEDITS++ and StyleAligned specialise in resolution-aware refinement, while TurboEdit and IP2P are two-stage pipelines that first predict a coarse edit mask. Blended diffusion performs training-free, CLIP-guided local edits via progressive noise-space blending for background preservation. FLUX.1 Kontext is a unified flow-matching in-context editor with fast multi-turn consistency. Nano Banana is a lightweight interactive editor focused on fast object and style edits with identity preservation. Evaluating against this diverse slate highlights TP-Blend's ability to blend rather than merely replace or stylise.

**Evaluation Protocol.** For our evaluation, we assembled a diverse set of high-resolution, publicly available images from Unsplash[1], following the same practice as prior work such as SLIDE Jampani et al. (2021) and Text-driven Image Editing via Learnable Regions Lin et al. (2024). The test dataset consists of 4,000 samples, created by pairing 40 base images with 20 distinct replace-blend object combinations and 5 distinct blend styles.

**Evaluation Metrics.** We assess alignment between generated image $I_g$ and four textual prompts—original object $P_O$, replaced object $P_R$, blend object $P_B$, and style $P_S$—using CLIP similarity:

$$\text{CLIP}_x = \cos\big(f_{\text{vis}}(I_g),\, f_{\text{text}}(P_x)\big), \quad x \in \{O, R, B, S\}.$$

Perceptual fidelity is quantified as $1-\text{LPIPS}_O$. To ensure comparability, each score $s$ is min-max normalized:

---

[1] https://unsplash.com/

Table 2: Performance on the full Object Replacement + Object & Style Blending task.

| Method | BOSM↑ | CLIP$_R$↑ | CLIP$_B$↑ | CLIP$_S$↑ |
|---|---|---|---|---|
| IP2P Brooks et al. (2023) (CVPR 2023) | 0.1277 | 0.1680 | 0.2776 | 0.1694 |
| LEDITS++ Brack et al. (2024) (CVPR 2024) | 0.2693 | 0.2039 | 0.2876 | 0.2236 |
| TurboEdit Deutch et al. (2024) | 0.3829 | 0.1954 | 0.2820 | 0.2090 |
| FLUX.1 Kontext Batifol et al. (2025) | 0.3955 | 0.2069 | 0.2942 | 0.2118 |
| StyleAligned Hertz et al. (2024) (CVPR 2024) | 0.4125 | 0.1888 | 0.2915 | 0.1973 |
| SeedEdit Shi et al. (2024) | 0.4650 | 0.1963 | 0.2915 | 0.2017 |
| Step1X-Edit Liu et al. (2025d) | 0.4652 | 0.2145 | 0.2920 | 0.2170 |
| Blended diffusion Avrahami et al. (2022) (CVPR 2022) | 0.4903 | 0.2096 | 0.2900 | 0.1805 |
| Nano Banana | 0.5849 | 0.2346 | 0.2856 | 0.2150 |
| **CAOF** | **0.6639** | 0.2014 | 0.2937 | 0.1976 |
| **CAOF+SASF** | **0.8656** | 0.2178 | 0.3022 | 0.2161 |

$$\hat{s} = \epsilon + (1 - \epsilon)\,\frac{s - s_{\min}}{s_{\max} - s_{\min}}, \quad \epsilon = 0.1.$$

The normalized scores are $\hat{\mathrm{CLIP}}_R$, $\hat{\mathrm{CLIP}}_B$, $\hat{\mathrm{CLIP}}_S$, and $1 - \hat{\mathrm{LPIPS}}_O$.

**BOM** (Blending Object Metric) measures replacement and blending accuracy:

$$\mathrm{BOM} = \frac{w_R + w_B + w_L}{\frac{w_R}{\hat{\mathrm{CLIP}}_R} + \frac{w_B}{\hat{\mathrm{CLIP}}_B} + \frac{w_L}{1 - \hat{\mathrm{LPIPS}}_O}},$$

**BOSM** (Blending Object Style Metric) further incorporates style fidelity:

$$\mathrm{BOSM} = \frac{w_R + w_B + w_S}{\frac{w_R}{\hat{\mathrm{CLIP}}_R} + \frac{w_B}{\hat{\mathrm{CLIP}}_B} + \frac{w_S}{\hat{\mathrm{CLIP}}_S} + \frac{w_L}{1 - \hat{\mathrm{LPIPS}}_O}}.$$

Both metrics are harmonic means where low individual scores significantly lower the final value, highlighting edits that successfully balance content fidelity and stylistic integration.

### 4.2 Comparisons with SOTA models

**Quantitative Evaluation of Object Replacement and Blending.** Table 1 presents BOM scores for 800 replacement–blend pairs. CAOF achieves the highest value (0.8388), substantially surpassing the next best method (0.7352). Its advantage does not stem from a single component: although Step1X-Edit yields the best CLIP$_R$ and SeedEdit tops CLIP$_B$, those gains are offset by weaker performance on the complementary cue and by larger perceptual drift, which the harmonic mean penalises. CAOF instead secures near-peak values on both alignment terms while also delivering the strongest image-fidelity score ($1 - \mathrm{LPIPS}_O = 0.8292$). This balance arises from the cost-aware transport in CAOF, which places blend features only at semantically consistent locations, preserving global structure and avoiding the artefacts or concept omission observed in the baselines. The results confirm that effective object blending requires simultaneous optimisation of replacement accuracy, blend consistency, and photographic integrity, a trade-off that CAOF best satisfies among the methods we evaluated.

**Quantitative Evaluation of Object Replacement, Blending, and Style Integration.** Table 2 lists all methods in ascending BOSM order. The lower half of the table shows that aggressive stylisation or simplistic blending hurts semantic alignment, producing BOSM below 0.40. Middle-ranking approaches recover object fidelity yet still dilute style cues, so their overall balance remains limited. Pure CAOF moves into the upper tier by preserving both objects without increasing perceptual drift, yielding BOSM 0.7102. Adding SASF raises the score to 0.9244, the largest margin in the study. This improvement is not obtained by style similarity alone: CLIP$_R$ and CLIP$_B$ also climb, indicating that the high-frequency details injected by DSIN and the text-driven Key-Value substitution sharpen local structure and make both

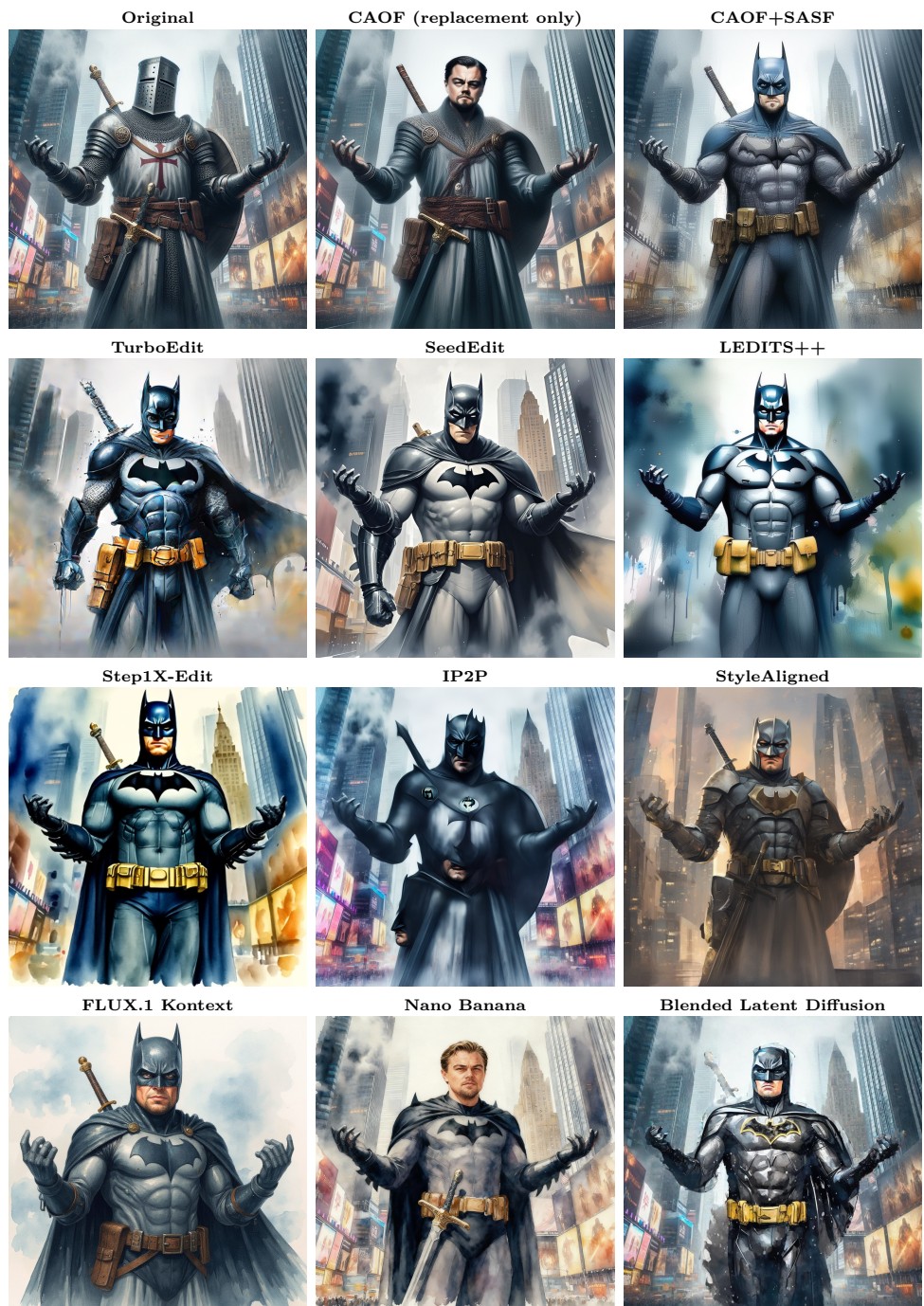

Figure 8: Method comparison for the task *Knight → Leonardo DiCaprio*, blended with *Batman* and rendered in a *water-color* style.

identities more recognisable. The joint optimisation of content and texture therefore proves essential when multiple conceptual constraints must be satisfied simultaneously.

**Visual Assessment.** Figure 8 (fantasy portrait) and Figure 9 (celebrity street scene) illustrate the quantitative trend reported in Table 2. CAOF+SASF achieves a balanced fusion where both the replaced identity,

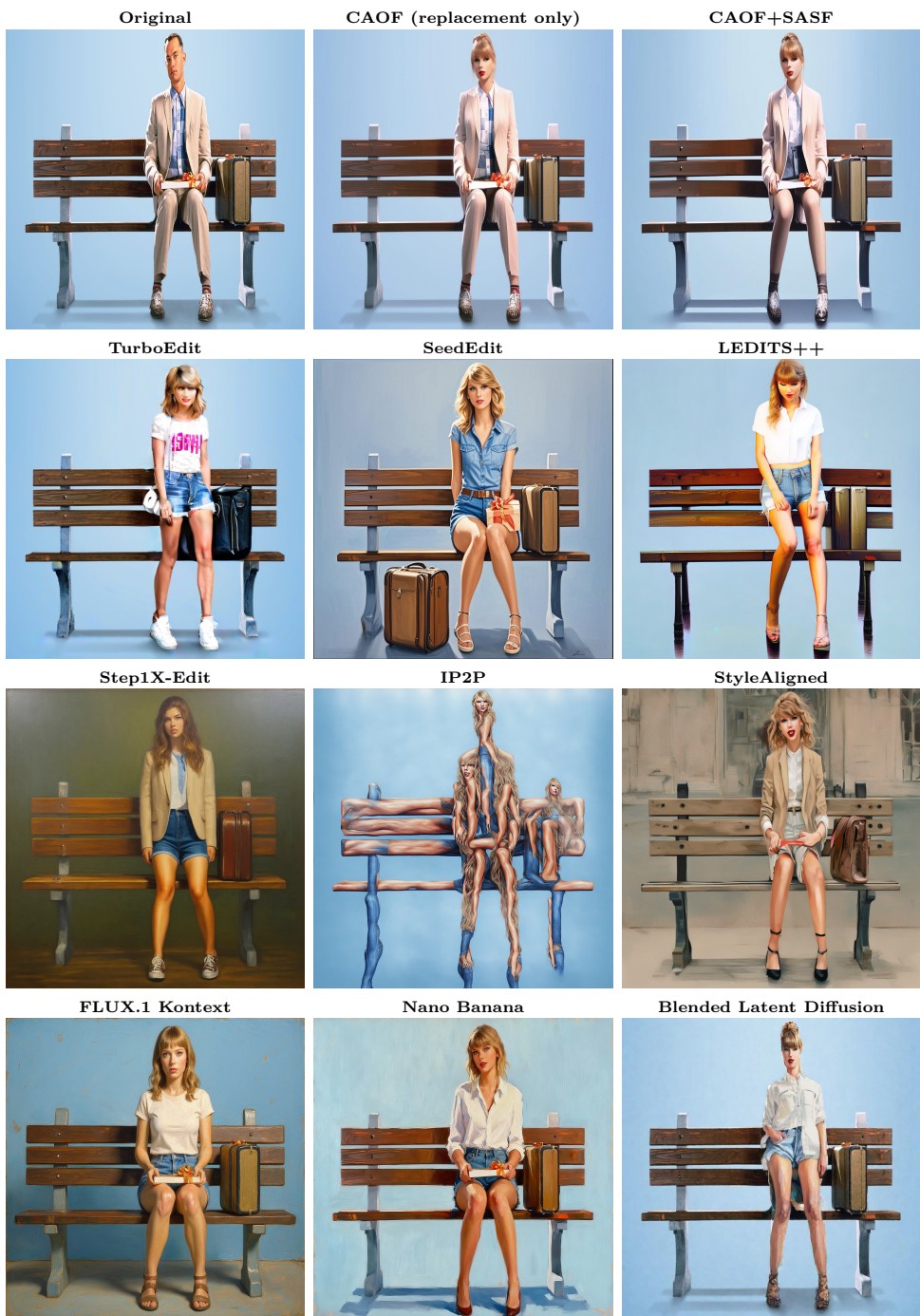

Figure 9: Method comparison for the task *Tom Hanks → Taylor Swift*, blended with *jean shorts+white shirt* and rendered in an *oil-painting* style.

the blended identity, and the target style are distinctly visible while maintaining the original scene geometry and background texture. In contrast, the baselines exhibit method-specific failure modes:

*Background degradation.* In Figure 8, StyleAligned Hertz et al. (2024) replaces the Times Square background with a different scene, and LEDITS++ Brack et al. (2024) and FLUX.1 Kontext Batifol et al. (2025) remove the background almost entirely. In Figure 9, StyleAligned Hertz et al. (2024), Step1X-Edit Liu et al. (2025d),

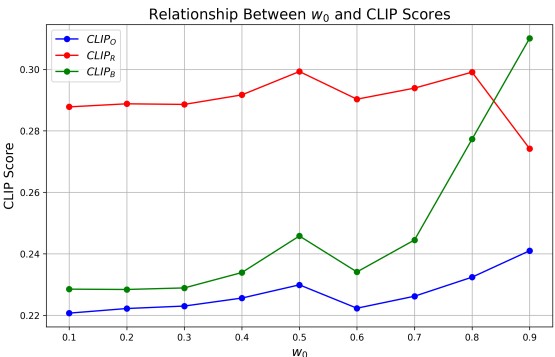

Figure 10: Variation of CLIP scores for original ($P_o$), replaced ($P_r$), and blend ($P_b$) object prompts as the blending coefficient $w_0$ changes. The curves illustrate CAOF's effectiveness in modulating blending strength, achieving the desired integration of the blend object while replacing the original object.

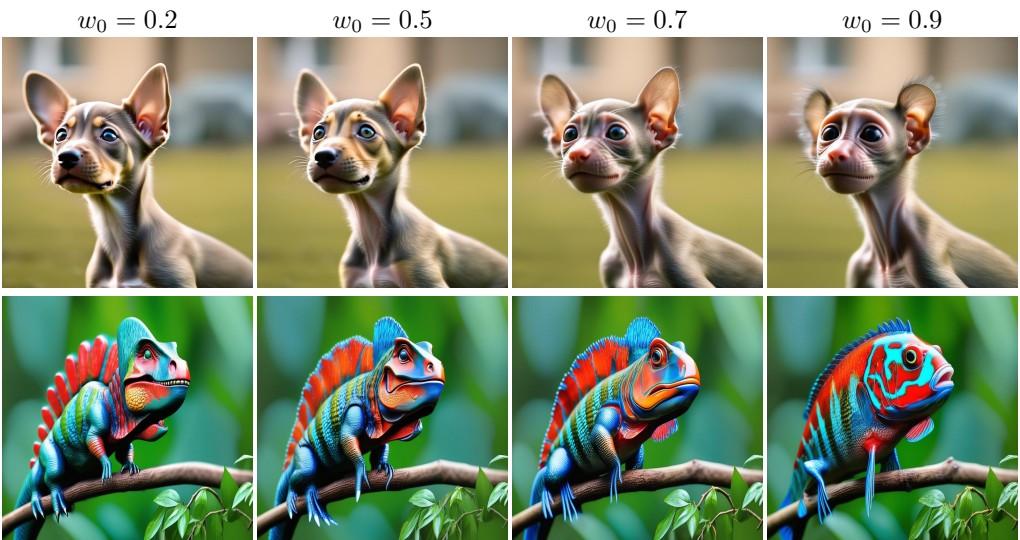

Figure 11: Object blending progression with varying blending coefficients $w_0$. Row 1: "**monkey**" blended into replaced "**puppy**" (originally "**alpaca**"). **Row 2**: "**fish**" blended into replaced "**dinosaur**" (originally "**chameleon**"). **Row 3**: "**ambulance**" blended into replaced "**jeep**" (originally "**truck**"). Row 4: "**Thanos**" blended into replaced "**knight**" (originally "**robot**"). Higher $w_0$ values correspond to increased blending intensity and finer textural details.

and FLUX.1 Kontext Batifol et al. (2025) substitute the original blue backdrop, while LEDITS++ Brack et al. (2024) partially destroys the bench. All baselines also drop secondary background details (e.g., leather shoes, socks, and the suit jacket) that are present in the source image.

*Loss of replaced-object identity after blending.* In Figure 8, TurboEdit Deutch et al. (2024), SeedEdit Shi et al. (2024), Step1X-Edit Liu et al. (2025d), IP2P Brooks et al. (2023), Blended diffusion Avrahami et al. (2022), and FLUX.1 Kontext Batifol et al. (2025) reduce or remove recognizable features of Leonardo DiCaprio. In Figure 9, the same methods fail to preserve identifiable traits of Taylor Swift under the jean-shorts + white-shirt blend.

*Severe distortions or unintended objects.* In Figure 8, SeedEdit introduces an extra arm, and IP2P generates duplicate faces. In Figure 9, SeedEdit adds an extraneous box, and IP2P produces a heavily distorted image with three faces. Across both figures, outputs from Blended diffusion Avrahami et al. (2022) are often coarse with blurred regions, obscuring small objects.

Table 3: OT ablation: CAOF vs. NoneOT.

| Method | BOM↑ | $\text{CLIP}_R$↑ | $\text{CLIP}_B$↑ | $1-\text{LPIPS}_O$↑ |
|---|---|---|---|---|
| NoneOT | 0.1429 | 0.1984 | 0.2891 | 0.8304 |
| **CAOF** | **0.2500** | 0.2014 | 0.2937 | 0.8292 |

Table 4: DSIN texture metrics versus $\alpha$ and $\sigma$.

| $\alpha$ | $\sigma$ | **LV**↑ | **GC**↑ | **HFS**↑ |
|---|---|---|---|---|
| 0.5 | 2.5 | 271.8709 | 79.9853 | $5.27\times10^9$ |
| 0.5 | 0.5 | 253.3316 | 79.7168 | $5.06\times10^9$ |
| 0.2 | 2.5 | 266.9800 | 80.0325 | $5.21\times10^9$ |
| 0.2 | 0.5 | 241.8779 | 76.5979 | $4.97\times10^9$ |
| 0.0 | – | 244.2984 | 68.8580 | $4.90\times10^9$ |

*Insufficient cross-identity blending.* In Figure 8, Nano Banana leaves the head largely unchanged as Leonardo DiCaprio, with little to no visible Batman attributes, indicating weak cross-identity fusion.

These issues, along with excessive denoising that washes out high-frequency details or spatial artifacts like duplicated limbs, result in lower BOSM scores for competing methods. This comparison highlights the perceptual advantage of CAOF+SASF, which maintains a coherent and natural fusion without introducing such distortions.

### 4.3 Ablation Study

**Ablation Study on CAOF.** To examine how CAOF controls the fusion strength, we vary the blending coefficient $w_0 \in [0.1, 0.9]$ (Eq. 9) and record the CLIP similarities for the original (O), replaced (R), and blend (B) prompts. Figure 10 illustrates the variation of CLIP scores with $w_0$. The curves clearly demonstrate CAOF's effectiveness in adjusting blending strength. As $w_0$ increases beyond 0.6, the influence of the blend object prompt $P_b$ significantly rises, while the influence of the replaced object prompt $P_r$ remains high until $w_0$ exceeds 0.8, after which it decreases rapidly. Concurrently, the influence of the original object prompt $P_o$ remains consistently low throughout, aligning with our goal to replace the original object with the replaced object while blending in the blend object to the desired extent. Qualitative frames in Fig. 11 corroborate the numerical trend, showing a smooth morph from "mostly replacement" to "mostly blend" without geometric break-down.

**SASF Ablation.** SASF relies solely on textual prompts for style specification, prompting us to measure style blending performance through $\hat{\text{CLIP}}_S$, the normalized similarity between the generated image $I_g$ and the style object prompt $P_s$. As shown in Table 2, CAOF+SASF attains a substantially higher $\text{CLIP}_S$ of 0.2161 than CAOF's 0.1976, indicating that SASF effectively injects the desired style features.

**Ablation on the joint percentile thresholds $\tau_{\text{source}}, \tau_{\text{dest}}$.** CAOF builds the source set $\mathcal{S}$ and destination set $\mathcal{D}$ by thresholding head-averaged cross-attention responses of the blend and replaced prompts. Positions above the $\tau_{\text{source}}$ percentile in the blend map enter $\mathcal{S}$ and those above the $\tau_{\text{dest}}$ percentile in the replaced map enter $\mathcal{D}$. Sweeping the *joint* threshold $\tau_{\text{source}} = \tau_{\text{dest}} \in \{0, 10, \ldots, 90, 99\}$ exposes a precision vs. coverage trade-off that directly governs how CAOF redistributes features. Figure 12 plots CLIP cosine scores for the original prompt $P_o$, the replaced prompt $P_r$, and the blend prompt $P_b$. Three observations follow from the measured curves.

First, $P_b$ peaks at a mid to high threshold: the best blend alignment occurs at 60% where $P_b = 0.2530$ and remains competitive at 70% with $P_b = 0.2371$. Around this regime spurious low-confidence tokens are removed yet all salient parts of the object remain in $\mathcal{S}$ and $\mathcal{D}$. The feature and spatial terms then rank candidate correspondences cleanly and the transport plan concentrates mass on semantically consistent matches, so the fused vectors carry the right identity and geometry.

Second, very high thresholds shrink coverage and favor replacement: when $\tau_{\text{source}}, \tau_{\text{dest}}$ are pushed to 80% and beyond, the sets collapse to a handful of extreme tokens. Coverage drops and CAOF touches only tiny regions, so the CFG-TE replacement signal dominates the denoising trajectory. Empirically $P_r$ rises from 0.1719 at 60% to 0.2361 at 99%, while $P_b$ falls sharply to 0.1592–0.1623.

Third, low thresholds dilute semantic precision: at 0%–50% the sets admit many background or off-object positions. The plan must spread mass across numerous weak matches and the averaged multi-head vectors

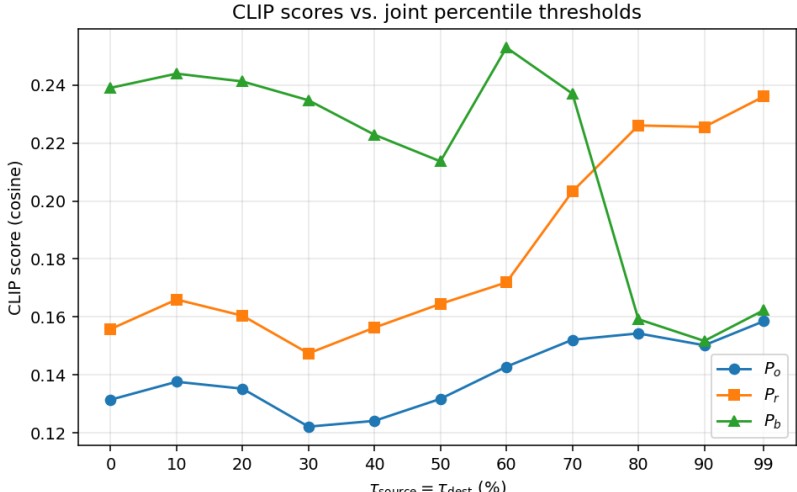

Figure 12: Joint percentile threshold ablation. CLIP cosine scores for $P_O$, $P_R$, and $P_B$ as the joint threshold $\tau_{\text{source}} = \tau_{\text{dest}}$ varies. The blend score $P_B$ peaks at 60%. Very high thresholds shrink $\mathcal{S}$ and $\mathcal{D}$ excessively and favor replacement ($P_R$) over blending, while low thresholds admit non-relevant vectors that dilute the fused signal. Moderate thresholds balance precision and coverage.

mix in non-relevant content, which reduces blend fidelity. This explains the drop from $P_b = 0.2413$ at 20% to $P_b = 0.2137$ at 50%.

Throughout the sweep the original content remains suppressed, with $P_o$ staying low in the range 0.1221–0.1585. Taken together these trends justify the default $\tau_{\text{source}} = \tau_{\text{dest}} \in \{0.6, 0.7\}$. This window removes noise while preserving spatial coverage, maximizes $P_b$ near its peak, keeps $P_r$ strong enough for reliable replacement and maintains low $P_o$. The ablation therefore confirms that mid to high joint percentiles are critical for stable and semantically faithful blending under CAOF.

**OT Ablation.** To disentangle the contribution of the Sinkhorn solver, we replace it with a naïve NONEOT variant that line-up source and destination tokens by index and applies a fixed $\alpha$-blend, thereby ignoring both feature similarity and spatial proximity. As summarised in Table 3, removing Optimal Transport slashes BOM from 0.2500 to 0.1429. The loss is driven almost entirely by lower alignment scores ($\text{CLIP}_R$ and $\text{CLIP}_B$), while the perceptual term $1 - \text{LPIPS}_O$ remains virtually unchanged. In other words, a uniform blend preserves low-level appearance but often allocates the wrong blend features to the wrong spatial regions, degrading semantic coherence. The cost-aware Sinkhorn plan redistributes those features toward geometrically and visually compatible destinations, yielding a markedly more faithful fusion without sacrificing overall image fidelity.

**DSIN Ablation.** Laplacian Variance (LV) Pertuz et al. (2013), GLCM Contrast (GC) Haralick et al. (1973), and FFT High-Frequency Sum (HFS) Gonzalez & Woods (2008) show that textural richness depends on the joint choice of the residual-mixing weight $\alpha$ and the Gaussian width $\sigma$, rather than on $\alpha$ alone. Raising $\alpha$ strengthens the amplitude of the injected high-frequency residual, but this extra energy is useful only if $\sigma$ is large enough to confine the smoothing kernel to genuinely low frequencies; with $\alpha = 0.5$ the wider kernel $\sigma = 2.5$ yields the highest LV, GC, and HFS, whereas the same $\alpha$ combined with the narrow kernel $\sigma = 0.5$ loses mid-range structure and drops all three scores. Conversely, keeping $\alpha$ moderate at 0.2 still improves over pure AdaIN ($\alpha = 0$), yet the gain is larger when $\sigma = 2.5$ than when $\sigma = 0.5$. These trends confirm that $\alpha$ governs how much fine detail is transferred while $\sigma$ sets the frequency band that will be regarded as "detail"; optimal texture emerges when both parameters are tuned together, explaining the peak at $\alpha = 0.5$, $\sigma = 2.5$ in Table 4 and the visibly crisper result in Figure 13.

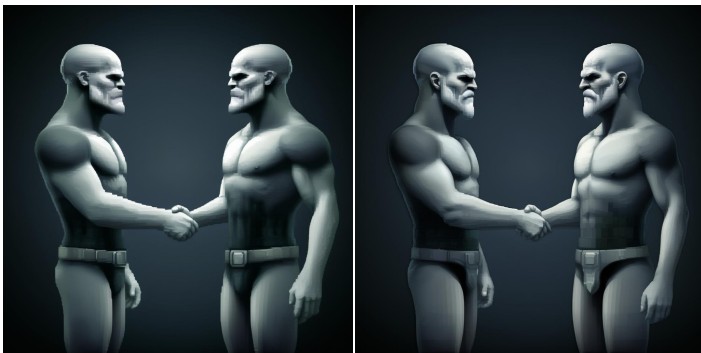

Figure 13: Pixel-art edit of "robot → knight" blended with "Thanos". Left: $\alpha = 0$, right: $\alpha = 0.5$, $\sigma = 2.5$.

# 5 Conclusion

We introduced TP-Blend, a training-free framework that performs object replacement, object blending, and style fusion within a single diffusion denoising run. By separating the content and style prompts, TP-Blend grants independent control over semantic structure and appearance. Cross-Attention Object Fusion employs an optimal-transport plan to place blend-object features in spatially and semantically consistent regions, while Self-Attention Style Fusion injects high-frequency texture through detail-sensitive instance normalisation and text-driven key-value substitution. Across extensive benchmarks, TP-Blend delivers sharper textures, stronger alignment with target objects and styles, and higher perceptual fidelity than recent editors, all without extra training or model fine-tuning. These results establish TP-Blend as a simple yet effective tool for precise, text-guided image editing within diffusion models.

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
