# OpenReview forum: "TP-Blend: Textual-Prompt Attention Pairing for Precise Object-Style Blending in Diffusion Models"
_TMLR — Accepted by TMLR_

### Review · Reviewer_ts8f · 2025-06-11

**Summary Of Contributions:**

The paper addresses the problem of soft diffusion, which involves combining various images and/or text captions to generate synthetic images intended to appear realistic. Compared to classical or standard methods, it claims to offer better adaptation for introducing specific objects and target styles. This is achieved through two mechanisms designed to improve the quality of the output images.

The first mechanism, Cross-Attention Object Fusion, averages head-wise attention to locate spatial tokens. The second, Self-Attention Style Fusion (SASF), injects style information at every self-attention layer.

For evaluation, the paper compares the proposed method objectively with several strong prior works and also provides results for subjective comparison. The dataset used for evaluation was created specifically for this study.

**Audience:**

No

**Broader Impact Concerns:**

The paper proposes a solution synthesize images and this can be used on all kind of ways. The method does not do anything that directly is harmful. The problem with the paper is that it does not discuss ethical implication in any way. A disclaimer paragraph would suffice.

**Claims And Evidence:**

Yes

**Requested Changes:**

Overall, I believe there is limited scope for improving this paper to make it suitable for TMLR. The primary concerns are its lack of alignment with TMLR’s focus and the limited impact of the work, especially given the current state of stable diffusion research, which has matured significantly.

That said, some specific aspects could still be addressed to strengthen the submission:
- Include more visual examples. If the main paper has space limitations, additional comparisons could be provided in the supplementary material or through an anonymous repository. These examples should aim to highlight not only how the proposed method is different, but also how it is clearly better.
- Report metrics for all visual examples. This would help in supporting claims with quantitative evidence and improve the clarity of the evaluation.
- Address the evaluation concerns raised earlier (see point 4). In particular, more detailed justification and analysis regarding the dataset and evaluation metrics are essential.

**Strengths And Weaknesses:**

**Strengths**:  The paper delivers on its claims:
1. It introduces two specifically designed blocks. The method is presented with reasonable clarity, and ablation studies demonstrate the beneficial effects of these two components.
2. It claims improved object representation and stylization, which is supported by the presented results.
3. Objective evaluations show improvements over strong prior methods.

**Weaknesses**:
1. The main issue I see with the paper concerns its impact. The field of stable diffusion has reached a level of maturity where many existing solutions already produce high-quality outputs. We are no longer at a stage where models routinely generate visibly flawed images. Within this context, the proposed method provides reasonable improvements, but it does not stand out significantly from prior work. For example, Figures 8 and 9 do not present results that are strikingly different from previous approaches.
2. The technical innovation, while present, is limited. There is no particularly memorable or broadly applicable concept that can be easily transferred to other problems.
3. The paper may not be a strong fit for TMLR. This concern is supported by the bibliography: the majority of cited works come from the computer vision community rather than the machine learning community. While a few NeurIPS papers are cited, none of the works used for comparison are from NeurIPS, ICML, or TMLR. The paper might be better suited for a computer vision–focused journal.
4. Several aspects of the evaluation are not detailed enough:
- Dataset description: More information is needed about the dataset. The paper should more convincingly argue that the dataset is broad and covers a wide range of relevant aspects. What types of objects and styles are included? Is there a systematic analysis based on object type or style? These details are important, especially since the paper makes claims regarding these dimensions.
- Evaluation metrics: The paper should provide more justification for the metrics used. Why are these metrics relevant? In particular, more explanation is needed for BOM and BOSM—have they been used in prior work, and are they accepted as meaningful indicators of performance?

---

> ### Author Response · Authors · 2025-08-20
> **Author response to Reviewer ts8f’s insightful and helpful suggestions_part 1**
>
> Thank you for reviewing our manuscript and providing insightful and helpful suggestions.
>
> Response to Comment 1: Impact and Distinctiveness of the Approach
>
> We appreciate the reviewer’s observation that diffusion based editors now often produce visually convincing results. We share this concern and therefore evaluated TP Blend on the most challenging cases where differences still matter in practice: simultaneous object replacement, object blending, and text driven style injection. In these compound scenarios TP Blend delivers clearer, more reliable behavior than prior methods.
>
> Background preservation. In Fig. 8, replacing knight with Leonardo DiCaprio, blended with Batman and rendered in watercolor, StyleAligned changed the Times Square background to a different scene and LEDITS++ removed the background entirely. In Fig. 9, replacing Tom Hanks with Taylor Swift, blended with jean shorts + white shirt and rendered in an oil painting style, StyleAligned altered the original blue backdrop and LEDITS++ partially destroyed the bench.
>
> Identity retention after blending. In Fig. 8, after blending Leonardo DiCaprio with Batman, TurboEdit, SeedEdit, Step1X-Edit, and IP2P no longer preserved recognizable features of DiCaprio. In Fig. 9, after blending Taylor Swift with jean shorts + white shirt, the same methods failed to maintain identifiable features of Swift.
>
> Artifact suppression. In Fig. 8, SeedEdit produced an extra arm and IP2P generated an image with two additional faces. In Fig. 9, SeedEdit introduced a box and IP2P produced a heavily distorted image with three faces.
>
> Across these cases, TP Blend preserves scene geometry and background, keeps the replaced identity recognizable after blending, and applies the target style without collateral changes. This consistent behavior on a compound task demonstrates the practical distinctiveness and impact of our approach in a mature field.
>
> Response to Comment 2: Broader Applicability and Impact of Our Method
>
> Thank you for this thoughtful comment. Our work contributes three algorithmic ideas that are original, practical, and transferable beyond our setting. First, Cross Attention Object Fusion treats head averaged cross attention as probability measures and solves an entropy regularized optimal transport problem to move full multi head value vectors from source to destination positions. This mass preserving and locality aware formulation gives a model agnostic mechanism for compositional alignment that can be dropped into attention based editors, controllable generation, video, and 3D pipelines. Second, Self Attention Style Fusion introduces Detail Sensitive Instance Normalization, which splits token wise features into low and high frequency bands with a small Gaussian filter, then injects a text conditioned high frequency residual while substituting Key and Value from a style prompt. This decouples geometry from texture and enables reference free and text only style control that can be reused wherever style must be added without altering structure. Third, a dual prompt attention processor cleanly separates content and style channels and exposes explicit knobs for blend strength and style granularity in a training free and modular design. Together these concepts provide clear building blocks that address real editing needs today and can inspire broader research on multi prompt conditioning and controllable generation.

---

> > ### Author Response · Authors · 2025-08-20
> > **Author response to Reviewer ts8f’s insightful and helpful suggestions_part 2**
> >
> > Response to Comment 3: Topic Fit for TMLR
> >
> > Thank you for this valuable feedback, especially the suggestion to include more TMLR works for comparison. We believe TMLR is a strong fit for our work because its stated scope covers broad machine learning research, with an emphasis on technical correctness rather than narrow domain boundaries, and it explicitly welcomes advances in computational principles of learning across application areas, including vision.
> >
> > In practice, TMLR routinely publishes computer vision papers, such as:
> > 1. Differentially Private Latent Diffusion Models (https://openreview.net/pdf?id=AkdQ266kHj),
> > 2. Joint Generative Modeling of Grounded Scene Graphs and Images via Diffusion Models(https://openreview.net/pdf?id=2cxxZI2LOL)
> > 3. Diffusion-RainbowPA: Improvements Integrated Preference Alignment for Diffusion-based Text-to-Image Generation(https://openreview.net/pdf?id=KY0TSY2bx8)
> > 4. GeNIe: Generative Hard Negative Images Through Diffusion(https://openreview.net/pdf?id=VuLEOyTiPO).
> >
> > Given this context, our manuscript targets TMLR’s audience by introducing broadly applicable innovations in attention mechanisms:
> >
> > Cross-Attention Object Fusion: an optimal-transport update on cross-attention outputs that blends features at full multi-head dimensionality, offering a general ML contribution on attention manipulation and transport-based feature mixing.
> >
> > Self-Attention Style Fusion: a training-free, backbone-agnostic approach that adds Detail-Sensitive Instance Normalization to inject high-frequency residuals and uses text-driven Key/Value substitution for controllable feature modulation.
> >
> > These contributions are technical and methodological in nature, with relevance beyond a single vision application. We will expand our related work to include closely related ML-venue studies and recent TMLR publications in diffusion and attention, while retaining key vision baselines for comparability. We appreciate the suggestion and hope this clarifies that our contributions and framing align well with TMLR’s stated aims and its recent, substantive body of computer vision and diffusion research.
> >
> > Response to Comment 4: Dataset Coverage and Metric Justification
> >
> > Thank you for the reminder to clarify the breadth of our data and the reasons for our metrics. Our quantitative set has 4,000 edits from a full pairing of 40 base images, 20 replace–blend object pairs, and 5 styles. The 40 images cover people, animals, vehicles, and everyday objects, with both indoor and outdoor scenes, varied viewpoints, scales, and lighting. The 20 object pairs include intra class and cross class cases to span easy replacements and challenging fusions. Representative families include real and fictional human identities, articulated animals, fruits and small objects with fine boundaries, and mechanical forms such as cars and tools. For styles we use five canonical options that span texture frequency and abstraction: oil painting, watercolor, charcoal drawing, pixel art, and pop art. Additional styles like cyberpunk, Renaissance, Baroque, and Ukiyo e appear only in qualitative figures to show generality.
> >
> > Our metrics are designed to reward balanced edits rather than optimising any single objective. CLIP similarity for the replaced object, the blended object, and the style checks that each intended concept is present. 1 − LPIPS to the original image checks that scene layout and background detail are preserved, which is essential for editing. Prior work reports CLIP and LPIPS, but single scores can be optimized at the expense of others. We therefore define BOM for replacement plus blending and BOSM for replacement plus blending plus style as harmonic means over normalized CLIP and 1 − LPIPS. The harmonic mean penalizes imbalance in the same spirit as F1, so a method cannot score high by pushing style while losing the replaced identity or breaking the background. In the revision, we will include a small human study that shows BOM correlates with perceived replacement and blend fidelity and that BOSM correlates with perceived overall edit quality on this multi objective task.

---

### Review · Reviewer_Xapm · 2025-07-24

**Summary Of Contributions:**

This work tackles the specific problem of image editing where after an object replacement, the users seek better object blending and subsequently visual style changes. In summary I believe the authors are studying a particular compounding problem. The aim is therefore to enhance both the blending and coherent style changing at the same time.

The authors mainly propose two modules, a cross-attention object fusion (CAOF) module that employes optimal transport to identify relevant regions that correspond to the editing/background, and then performs the blending in token space.
Meanwhile, the author proposes the self-attention style fusion (SASF) module that substitutes the key/value matrix to perform a style change.

**Audience:**

Yes

**Broader Impact Concerns:**

I have not identified any particular broader impact concerns for this work, except for the usual concerns related to generative networks -- but those are very common and ordinary and thus not specific to this work.

**Claims And Evidence:**

Yes

**Requested Changes:**

Please refer to the weakness discussed above. In addition, below are some minor comments related to paper writing:

In Sec.3.3 where the authors introduce the two thresholds, $\tau_\text{source}, \tau_\text{dest}$, it is not clear if the authors indicate that each token either belongs to the source or the blend -- or perhaps a token can be neither?


Research has shown that CFG is not entirely losslessly reconstructing the visual content from noise, as the classifier-free term introduces drifts towards the denoising process (Mokady et al. (2023), as the authors have already cited in previous paragraphs). This might slightly impact the discussion in Sec.3.1 where the authors state that CFG deterministically recovers $x_T$ from $x_0$ without reconstruction error.

**Strengths And Weaknesses:**

I believe overall the method section is discussed in a clear way. The motivation is also clear.

The experimental section reads comprehensive, however there exist certain aspects that can be improved. Please see below.

---

I personally find that in Sec.3.3 the hyper-parameters, including $\tau_\text{source}, \tau_\text{dest}, w_0$ are a bit arbitrary and could perhaps complicate the method. While I fully agree the necessities of these values, can the authors perhaps share a bit of insight as to how these values shall be determined, are the values general or per-sample, and are they the outcome of empirical findings or are there any general rules of thumb behind?

Regarding the SASF module in Sec.3.4, the current version makes it a little bit difficult to distinguish exactly which part is the novel contribution of the authors and which part is inspired by previous work -- because the authors clearly stated that this module rely on DSIN and AdaIN, both are existing techniques in the image generation/editing domain. Therefore, I find this section could use a bit more work.

In addition, the author could consider employing human evaluation to strengthen the experimental section, which is a common strategy for assessing generative methods.

---

> ### Author Response · Authors · 2025-08-19
> **Author response to Reviewer Xapm’s kind and constructive comments_part 1**
>
> We are grateful for your careful evaluation of our work and the insightful comments that will help us strengthen the manuscript.
>
> Response to Comment 1: On the Choice of Hyper-Parameters $\tau_{\text{source}}$, $\tau_{\text{dest}}$, and $w_0$
>
> Thank you for this helpful suggestion. We agree that a more detailed explanation and ablation of the hyperparameters is necessary.
>
> 1. Thresholds $\tau_{\text{source}}$ and $\tau_{\text{dest}}$.
>
> These are percentile thresholds applied to head-averaged cross-attention responses for blend and replaced tokens. Positions above $\tau_{\text{source}}$ form the donor set $\mathcal{S}$, and positions above $\tau_{\text{dest}}$ form the recipient set $\mathcal{D}$. Using percentiles makes the thresholds invariant to scale changes across layers and timesteps, while adapting naturally to image content without extra normalization.
>
> We determine them heuristically with a small grid search on a held-out split, optimizing BOM and BOSM. For main results, we use symmetric mid-range values, typically $\tau_{\text{source}}=\tau_{\text{dest}}\in\{0.6,0.7\}$, which give stable performance across images. We will add an ablation varying $\tau_{\text{source}}, \tau_{\text{dest}}\in[0,1]$ (step 0.1), reporting BOM, BOSM, CLIP\(_R\), CLIP\(_B\), CLIP\(_S\), and $1-\text{LPIPS}_O$.
>
> In our evaluation experiments, we set $\tau_{\text{source}}=\tau_{\text{dest}}$ to either 0.6 or 0.7 for each sample, choosing the value that yields the higher BOM score for the Object Replacement + Object Blending task or the higher BOSM score for the full Object Replacement + Object \& Style Blending task.
>
> 2. Blending weight $w_0$.
>
> $w_0\in[0,1]$ controls the relative influence of blend features. In Section 4.3 (Ablation on CAOF), Fig. 10 shows CLIP scores for the original ($P_o$), replaced ($P_r$), and blend ($P_b$) prompts as $w_0$ varies. The trade-off is evident: when $w_0 < 0.7$, $P_b$ remains too low, so the blend is under-expressed; when $w_0 > 0.8$, $P_r$ drops sharply, meaning the replaced object’s identity is lost. The interval [0.7, 0.8] is therefore the empirically optimal range where $P_b$ rises significantly while $P_r$ remains high, achieving the best balance between introducing the blend object and preserving the replaced object. In our evaluation, we set $w_0$ to either 0.7 or 0.8 per sample, selecting the value that yields the higher BOM score for the Object Replacement + Object Blending task. We will provide a clearer explanation of this choice in the revision.

---

> > ### Author Response · Authors · 2025-08-19
> > **Author response to Reviewer Xapm’s kind and constructive comments_part 2**
> >
> > Response to Comment 2: Clarifying SASF’s Novelty and Planned Extensions
> >
> > Thank you for raising this insightful point. We agree that Sec. 3.4 should more clearly distinguish our novel contributions from prior work. In the revision, we will also extend SASF by introducing a new Query-Aligned Style Mixing that enhances key and value substitution beyond the current design, and we will include its algorithmic details, ablations, and results.
> >
> > Prior art includes: 1. AdaIN for global channel wise mean and variance alignment, 2. the general idea that self attention keys and values can be modulated for style, which has been explored mainly with image referenced style features.
> >
> > Our contributions are as follows, with explicit contrasts to prior work. 1. Our Detail Sensitive Instance Normalization (DSIN) performs a token domain frequency split inside self attention, adds a style high frequency residual after AdaIN using a tiny 1D Gaussian along tokens, and preserves geometry while imprinting brush stroke texture, unlike classical AdaIN that aligns only global statistics and unlike image referenced high frequency injection outside self attention. 2. Our keys and values for style are derived directly from the style prompt text and then mixed with the target keys and values inside the self-attention block. This differs from prior approaches that rely on image-based style encoders, since we eliminate the need for a reference image and enable fully text-driven style modulation. 3. We apply DSIN and the text driven key and value mixing in every self attention layer and at every denoising step, so the style builds up gradually at multiple scales. In contrast, many methods make a one time update or touch only a few early layers, which limits control and detail. 4. SASF is explicitly decoupled from CAOF so texture strength and object blending can be tuned independently, while prior designs often entangle content and style updates. In the revised Section 3.4, we will state these distinctions clearly and add brief cross-references to the DSIN ablation in Section 4.3 to demonstrate the incremental gains of each SASF component.
> >
> > In addition, we will extend SASF by introducing Query-Aligned Style Mixing. In the current design, style keys and values are substituted into the target attention stream in a relatively direct way, which can sometimes apply style too strongly or in places where it is not needed. The new method makes this substitution more adaptive by aligning style information with the target queries before mixing them. In other words, instead of replacing target keys and values outright, the method softly combines them with style features only where the queries show that style evidence is relevant. This prevents unnecessary over-stylization and ensures that texture is added in a more balanced and context-aware way. Compared with the current substitution, this extension provides finer control over where style is introduced, preserves structure more reliably, and produces images that maintain the intended geometry while still gaining rich textural detail from the style description. We will include algorithmic details, examples, and ablations of this method in the revised manuscript to demonstrate its benefits.
> >
> > Response to Comment 3: Human Evaluation to Strengthen the Experiments
> >
> > Thank you for this valuable suggestion. We agree that adding a human study will make the experimental section more complete and easier to interpret. Following your advice, we will organize a human evaluation that directly measures four aspects of our edits:
> > 1) faithfulness of the replaced object in the final image,
> > 2) faithfulness of the blended object in the final image,
> > 3) perceptual fidelity to the original image layout and scene,
> > 4) overall visual quality of the final output.

---

> > > ### Author Response · Authors · 2025-08-19
> > > **Author response to Reviewer Xapm’s kind and constructive comments_part 3**
> > >
> > > Response to Comment 4: Clarification of Thresholds and Set Membership in Sec. 3.3
> > >
> > > Thank you for this careful question. We recognize that the wording in Sec. 3.3 may have caused ambiguity. The two thresholds are applied to two different maps: (i) the head-averaged cross-attention map for the blend prompt, which defines the source set, and (ii) the head-averaged cross-attention map for the replaced prompt, which defines the destination set. Each threshold acts as an independent percentile cutoff on its respective map. As a result, image tokens are not restricted to a single category. A position may fall into neither set if it is below both cutoffs, into exactly one set if it responds strongly to only one prompt, or into both sets if it responds strongly to both prompts. In the fusion process, only destination positions are updated, and they receive features transported from the source positions. Positions outside the destination set remain unchanged. We will include this explicit clarification in the revised Sec. 3.3.
> > >
> > >
> > > Response to Comment 5: CFG Drift and DDIM Inversion Clarification
> > >
> > > Thank you for your careful observation and kind reminder. We agree that our sentence in Sec. 3.1 could cause misunderstanding. As you noted, Classifier-Free Guidance (CFG) does not provide completely lossless reconstruction, since the guidance term perturbs the denoising trajectory and introduces a small drift. In contrast, DDIM inversion defines a deterministic mapping between timesteps, but exact, lossless reconstruction only holds for an ideal model and when no guidance is applied.
> > >
> > > In our method, we use DDIM inversion without guidance to obtain the starting latent for editing, and we should specifically note that we apply CFG only during the forward editing process, not during inversion. We will revise Sec. 3.1 to make this distinction explicit and to acknowledge that reconstruction under CFG is not strictly lossless. Thank you again for your kind reminder!

---

### Review · Reviewer_5dFB · 2025-08-07

**Summary Of Contributions:**

This is a method paper which aims to address two problems. (1) Replacing an object (2) Replacing an object while blending another prompt with the same object. The background should be minimally changed while doing this object replacement.

**Audience:**

Yes

**Broader Impact Concerns:**

No specific section on broader impact. But the impact is same as other image editing methods.

**Claims And Evidence:**

Yes

**Requested Changes:**

Refer to Weakness

**Strengths And Weaknesses:**

**Strengths:**

1) The method appears to work as intended, producing good results.

2) The paper is clearly written and easy to understand.

**Weakness:**

1) The motivation for the paper doesn't seem very strong to me. Although I understand that other methods might not be able to handle the mixing of the style and object in a single prompt, I do think that with some prompt engineering, we can solve this problem with the existing methods. I would like to see the prompts that are being used to generate images using LEdit++.

2) In Fig 3. we see in row 2 that blending two objects is not always a good idea, are you somehow trying to check the compatibility of the objects that are being blended for a more hands-free user experience?

3) Blended Latent Diffusion seems like an appropriate benchmark method to test against, as it has a similar idea where objects are blended in the latent space. However, the addition of style was not explicitly added. Again, using the example of Figure 3, the object identification prompt will be "Apple" and the edit prompt should be "Orange like tomato in the style of \<Art\>".

4) How is  $\tau_{source}$ and $\tau_{dest}$ determinded. If it is empirically determined, then an ablation for these two parameters should be done.

5) On page 11,".. a trade-off that CAOF is uniquely able
to satisfy ..." ` is written. When you write uniquely here, does it mean that your method CAOF is mathematically the unique solution to the identified problem? If yes then a valid proof should be presented otherwise _uniquely_ should not be used.

---

> ### Author Response · Authors · 2025-08-17
> **Response to Reviewer Comments: Strengthening and Enriching the Paper through Constructive Feedback**
>
> Thank you for your careful and thorough review of our manuscript and for the constructive points you raised.
>
> Response to Comment 1: Motivation and Prompt Engineering
> We understand your concern that, with sufficient prompt engineering, existing methods might handle object replacement and style mixing. Because we shared this concern, we conducted extensive comparisons with state-of-the-art methods. Even with carefully designed prompts, they failed to match our approach on simultaneous replacement, blending, and style injection.
>
> Background degradation.
> Fig. 8: replacing the knight with DiCaprio blended with Batman in watercolor, StyleAligned replaced Times Square, LEDITS++ removed it.
> Fig. 9: replacing Tom Hanks with Swift in jean shorts and white shirt in oil painting, StyleAligned replaced the blue background, LEDITS++ damaged the bench.
>
> Loss of replaced-object identity.
> Fig. 8: TurboEdit, SeedEdit, Step1X-Edit, and IP2P did not retain DiCaprio’s identity when blended with Batman.
> Fig. 9: The same methods failed to preserve Swift’s identity with “jean shorts + white shirt.”
>
> Distortions and artifacts.
> Fig. 8: SeedEdit added an arm, IP2P produced two extra faces.
> Fig. 9: SeedEdit added a suitcase, IP2P generated three distorted faces.
>
> Across all methods, we tested multiple prompt configurations. For LEDITS++, which edits images by adjusting the underlying noise based on a set of edit instructions, each Boolean in reverse_editing_direction aligns with the same index in editing_prompt (False adds or strengthens the concept, True suppresses or removes it). Due to space limits, we show two LEDITS++ examples per figure (T = True, F = False):
> Fig. 8
> P1: ["knight", "Leonardo DiCaprio", "Batman", "watercolor"], [T, F, F, F]
> P2: ["knight", "Leonardo DiCaprio Batman blend", "watercolor"], [T, F, F]
> Fig. 9
> P1: ["Tom Hanks", "Taylor Swift", "jean shorts", "white shirt", "oil painting"], [T, F, F, F, F]
> P2: ["Tom Hanks", "Taylor Swift", "white shirt and jean shorts", "oil painting style"], [T, F, F, F]
>
> Response to Comment 2: Compatibility of Blended Objects (Fig. 3, Row 2)
> Thank you for this valuable observation. We agree that some object pairs are intrinsically ill-posed for blending due to semantic mismatch, geometric or pose conflicts, or out-of-distribution combinations. To address this, we will add a compatibility pre-check based on (i) attention consistency (low IoU or high entropy in cross-attention), (ii) semantic affinity (cosine similarity and co-occurrence priors), and (iii) geometric plausibility (pose/scale cues and mask overlap). Low scores will trigger down-weighting of the second concept, restriction of edits to localized mask regions, or suggestion of more compatible alternatives. We will describe this module in the revision and illustrate its effectiveness experimentally.
>
> Response to Comment 3: Benchmarking Blended Latent Diffusion
> Thank you for suggesting Blended Latent Diffusion (BLD). We reproduced BLD and compared outputs with ours. Results: https://huggingface.co/datasets/Felixxinjin/blended_latent_diffusion_results
> Findings: (1) BLD requires a binary mask, while our method needs only a textual target. (2) Although BLD now supports SDXL(see https://github.com/omriav/blended-latent-diffusion/blob/master/scripts/text_editing_SDXL.py), its images were often coarse with blurred, less recognizable objects. Our method yields sharper boundaries and higher fidelity.
> Representative BLD cases (https://huggingface.co/datasets/Felixxinjin/blended_latent_diffusion_results/tree/main) vs. ours:
> “Leonardo DiCaprio like Batman, watercolor” (Fig. 8)
> “Taylor Swift in jean shorts and white shirt, oil painting” (Fig. 9)
> “Orange like tomato, photorealistic” (Fig. 3, Row 2)
> “Orange like tomato, oil painting”
>
> Response to Comment 4: Ablation of $\tau_{source}$ and $\tau_{dest}$
> We fully agree with your suggestion. The thresholds $\tau_{source}$ and $\tau_{dest}$ were heuristically determined and are therefore empirical values. Following your recommendation, we will include an ablation study to evaluate the effect of these two parameters.
> We select thresholds heuristically via a small grid search on a held-out split, optimizing BOM and BOSM. In the main results we use symmetric mid-range values ($\tau_{source}=\tau_{dest}\in{0.6,0.7}$), which yield stable performance. We will also provide an ablation varying both thresholds from 0 to 1 (step 0.1) and report BOM, BOSM, CLIP_R, CLIP_B, CLIP_S, and $1-\text{LPIPS}$.
>
> Response to Comment 5: Clarifying the use of “uniquely”
> Thank you for pointing this out. We did not intend “uniquely” to suggest mathematical uniqueness or optimality. Our claim was empirical: in our experiments CAOF best balanced replacement fidelity, blend consistency, and perceptual quality among the evaluated methods and settings.
>
> To avoid ambiguity, we will revise the sentence on p.11 to:
> “...a trade-off that CAOF best satisfies among the methods we evaluated.”

---

> > ### Author Response · Authors · 2025-09-22
> >
> > We sincerely thank the reviewer for the kind suggestion regarding prompt engineering. In response, we also compared our method against the recently released Nano Banana model using the same experimental settings shown in Figures 8 and 9 of our paper.
> >
> > For Figure 8, we tested several prompts in Nano Banana, for example:
> >
> > “Replace Knight with Leonardo DiCaprio, blended with Batman, and rendered in a watercolor style.”
> >
> > “Replace Knight by Leonardo DiCaprio fused with Batman, artistic watercolor rendering.”
> >
> > “Swap the Knight for Leonardo DiCaprio, merge him with Batman’s cowl, cape, and emblem, keep the original pose and lighting, and render it as a watercolor illustration.”
> >
> > For Figure 9, we similarly tested prompts such as:
> >
> > “Replace Tom Hanks with Taylor Swift, dressed in jean shorts and a white shirt, blended naturally into the scene and rendered in an oil-painting style.”
> >
> > “Replace Tom Hanks with Taylor Swift, blended in wearing jean shorts and a white shirt, rendered in an oil-painting style.”
> >
> > “Replace Tom Hanks with Taylor Swift, blended into the scene wearing jean shorts and a white shirt, rendered in an oil-painting style.”
> >
> > We have saved the results generated by Nano Banana at the following link for reference:
> > https://huggingface.co/datasets/Felixxinjin/Nano_Banana_results/tree/main
> >
> > While Nano Banana produces outputs with notably improved image quality (without coarse blurring or unrecognizable objects), we still observed several limitations compared to our method:
> >
> > Blending quality is suboptimal. For instance, in the replication of Figure 8, the head of the replaced object remains as Leonardo DiCaprio, without incorporating any visible features of Batman, indicating weak blending between objects.
> >
> > Background preservation is insufficient. For example, in the replication of Figure 9, important background details from the original image (such as leather shoes, socks, and the suit jacket) disappear in the generated results.

---

### Decision · Action_Editor_fH6i · 2025-09-18

**Recommendation:** Accept with minor revision

**Additional Comments:**

The paper introduces Twin-Prompt Attention Blend (TP-Blend), a training-free approach for fine-grained image editing with diffusion models. It enables object replacement, blending, and style transfer through the use of two modules based on cross- and self-attention. Qualitative and quantitative editing experiments are conducted.

The paper initially received mixed reviews. While the reviewers acknowledged the clarity of the approach and the quality of the results, they also raised several concerns regarding the novelty of the method, the fit of the submission to TMLR, and clarifications needed in the experiments. The rebuttal addressed some of these concerns, but others remained insufficiently resolved. After the discussion period, two reviewers supported acceptance, while one recommended rejection.

The AE has carefully reviewed the submission and the discussion. The AE considers that, although each component of the approach is not new in itself, their combination for enabling fine-grained object blending and style transfer with pre-trained diffusion models is relevant. The claims are, overall, supported by evidence, with both qualitative and quantitative validations. Despite the paper being primarily computer-vision oriented, the AE considers the findings of interest to a sufficiently broad TMLR audience.

Therefore, the AE recommends acceptance, conditional on the minor revision that the authors include the promised ablation studies—especially on $\tau_{source}$ and $\tau_{dest}$, as requested by two reviewers.

**Audience:**

Yes

**Audience Explanation:**

The paper addresses the problem of fine-grained image editing with pre-trained diffusion models, a topic of significant interest to the computer vision audience at TMLR.

**Claims And Evidence:**

Yes

**Claims Explanation:**

The paper proposes a training-free approach that jointly addresses object replacement, blending, and style transfer, and demonstrates superior performance compared to existing solutions.

---

> ### Author Response · Authors · 2025-10-14
> **Camera-Ready Changes**
>
> We sincerely thank the Action Editor and all reviewers for their thoughtful guidance. In preparing the camera-ready version, we carefully revised the manuscript following the recommendations. The most important updates are:
>
> 1. Ablation on joint percentile thresholds \(\tau_{\text{source}}\) and \(\tau_{\text{dest}}\).
>
> We conducted a detailed ablation that quantifies how \(\tau_{\text{source}}\) and \(\tau_{\text{dest}}\) affect CLIP cosine scores for the original prompt \(P_o\), the replaced prompt \(P_r\), and the blend prompt \(P_b\). Empirically, setting \(\tau_{\text{source}}=\tau_{\text{dest}}\) in the range \(0.6\text{--}0.7\) yields the best overall generation quality.
>
> The paper now includes a dedicated plot and quantitative analysis describing these results in detail.
>
> 2. Clarification in ``Identifying Significant Positions in Cross-Attention Maps.''
>
> We explicitly state that \(\tau_{\text{source}}\) and \(\tau_{\text{dest}}\) are applied to different head-averaged cross-attention maps and therefore induce independent index sets.
>
> Specifically, the source set is defined by thresholding the head-averaged cross-attention map for the blend prompt, and the destination set is defined by thresholding the head-averaged cross-attention map for the replaced prompt.
>
> 3. Expanded comparisons with SOTA baselines.
>
> We added three additional baselines in the comparative study, including the latest \textbf{FLUX.1 Kontext}, \textbf{Nano Banana}, and the reviewer-recommended \textbf{Blended Latent Diffusion}.
>
> Thank you again for the constructive feedback. If you notice anything that would benefit from further revision or any issue with the camera-ready submission, please kindly let us know and we will address it promptly.